# Atmospheric and surface observations during the Saint John River Experiment on Cold Season Storms (SAJESS)

Hadleigh D. Thompson[1], Julie M. Thériault[1], Stephen J. Déry[2], Ronald E. Stewart[3], Dominique Boisvert[1], Lisa Rickard[2], Nicolas R. Leroux[1], Matteo Colli[5] and Vincent Vionnet[4]

[1]Department of Earth and Atmospheric Sciences, Centre ESCER, Université du Québec à Montréal, Montréal, Quebec, H3C 3P8, Canada
[2]Department of Geography, Earth and Environmental Sciences and Natural Resources and Environmental Studies Program, University of Northern British Columbia, Prince George, British Columbia, V2N 4Z9, Canada
[3]Department of Environment and Geography, University of Manitoba, Winnipeg, Manitoba, R3T 2N2, Canada
[4] Meteorological Research Division, Environment and Climate Change Canada, Dorval, Quebec, H9P 1J3, Canada
[5]Artys Srl, Piazza della Vittoria, 9/3, 16121 Genova, Italy

*Correspondence to*: Julie M. Thériault (theriault.julie@uqam.ca)

**Running title**

Data from the Saint John River Experiment on Cold Season Storms (SAJESS)

**Abstract.** The amount and phase of cold season precipitation accumulating in the upper Saint John River (SJR) basin are critical factors in determining spring runoff, ice-jams, and flooding. To study the impact of winter and spring storms on the snowpack in the upper SJR basin, the Saint John River Experiment on Cold Season Storms (SAJESS) was conducted during winter/spring 2020-21. Here, we provide an overview of the SAJESS study area, field campaign, and data collected. The upper SJR basin represents 41% of the entire SJR watershed and encompasses parts of the US state of Maine and the Canadian provinces of Quebec and New Brunswick. In early December 2020, meteorological instruments were co-located with an Environment and Climate Change Canada station near Edmundston, New Brunswick. This included a separate weather station for measuring standard meteorological variables, an optical disdrometer, and a micro rain radar. This instrumentation was augmented during an intensive observation period that also included upper-air soundings, surface weather observations, a multi-angle snowflake camera, and macrophotography of solid hydrometeors throughout March and April 2021. During the study, the region experienced a lower-than-average snowpack that peaked at ~65 cm, with a total of 287 mm of precipitation (liquid equivalent) falling between December 2020 and April 2021, a 21% lower amount of precipitation than the climatological normal. Observers were present for 13 storms during which they conducted 183 hours of precipitation observations and took more than 4000 images of hydrometeors. The inclusion of local volunteers and schools provided an additional 1700 measurements of precipitation amounts across the area. The resulting datasets are publicly available from the Federated Research Data Repository at https://doi.org/10.20383/103.0591 (Thompson et al., 2023). We also include a synopsis of the data management plan, and a brief assessment of the rewards and challenges of conducting the field campaign and utilizing community volunteers for citizen science.
.

## 1 Introduction

The Saint John River Experiment on Cold Season Storms (SAJESS) focused on cold region processes related to winter and spring storms over the transboundary upper Saint John River basin, located on the border of Maine (ME) and the provinces of Quebec (QC) and New Brunswick (NB). The Saint John River, known as the Wolastoq by local Indigenous communities, is 673 km long and drops 480 m in elevation from its source at the Little John River (ME) down to the Bay of Fundy (Fig. 1). The Saint John River watershed covers 55,000 km$^2$, with 36% located in the U.S., although, here we define the upper Saint John River basin as the area that drains into the Saint John River above Grand Falls, NB. Economically important to the region, the river provides flow for five hydroelectric facilities with development being overseen by the International Joint Commission (Kenny & Secord, 2010).

A concern of emergency managers along the Saint John River is the risk of catastrophic flooding when spring rain coincides with relatively high temperatures, creating significant snow melt. Such flooding events occurred in 2008, 2018, and again in 2019, and were in the annual top 10 Canadian weather disasters identified by Environment and Climate Change Canada (ECCC) (ECCC, 2017, 2019, 2020). Although this sub-catchment covers an area of 22,600 km$^2$, most of the research focusing on the Saint John River does so by examining the lower reaches and associated lakes, wetlands, and tidal estuaries. There is therefore a paucity of both hydrological knowledge of the upper basin, as noted by Budhathoki et al. (2022), and meteorological stations (only two within the upper SJR basin) (also see Fortin and Dubreuil, 2020).

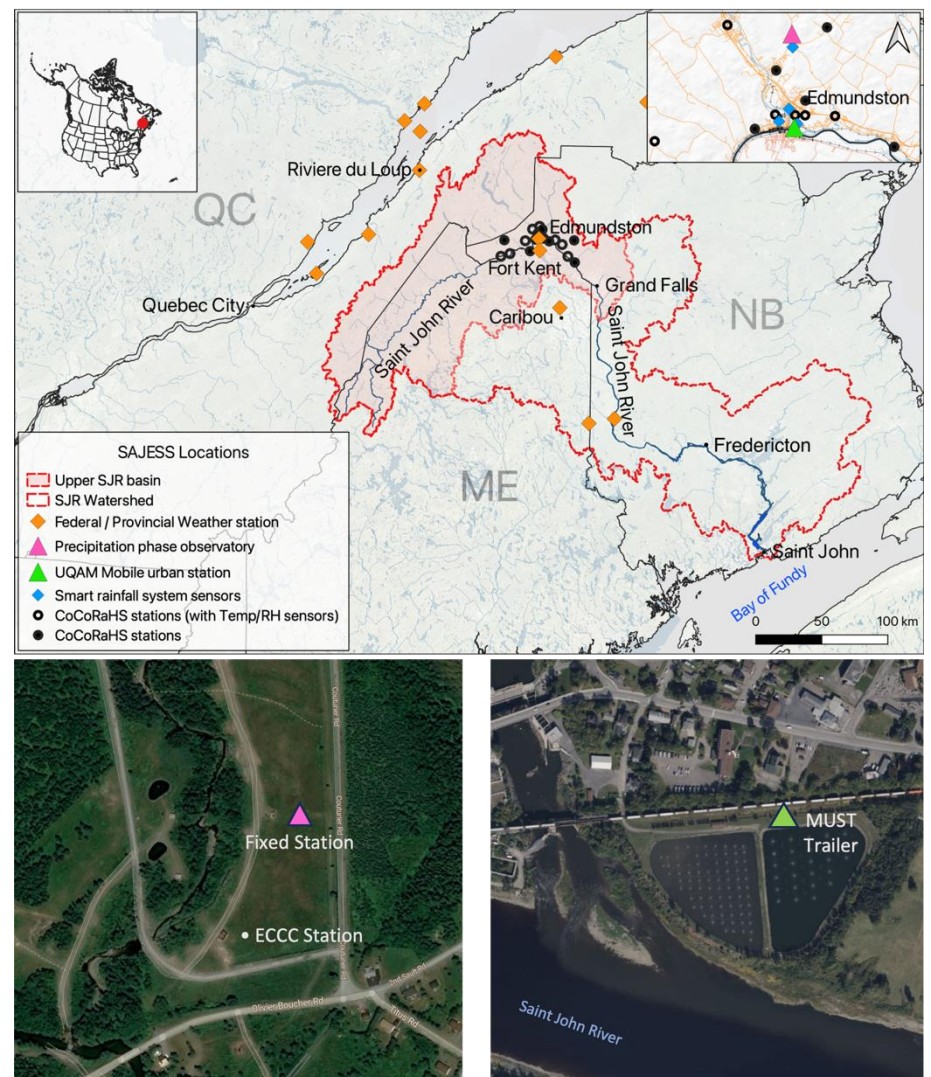

**Figure 1: Top: The Upper Saint John River Basin (shaded red), straddling the borders of Quebec (QC), Maine (ME, US) and New Brunswick (NB) is a sub-basin of the Saint John River Basin (red line). The Environment and Climate Change Canada (ECCC) and US National Weather Service (NWS) weather stations, SAJESS-supplied CoCoRaHS stations including where Temperature/RH sensors were also co-located, (black circles), the Precipitation Phase Observatory/Fixed Station (pink triangle) and the UQAM Mobile urban station/Must Trailer (green triangle) are shown. Bottom: Aerial photographs of the two main SAJESS sites showing the locations of the Fixed Station, MUST Trailer, and ECCC Station. Microsoft product screenshots reprinted with permission from Microsoft Corporation.**

There have been several previous cold-season precipitation studies in North America. These include a comparison between orographic winter storms in the San Juan Mountains and Sierra Nevada by Marwitz (1986), the Canadian Atlantic Storms Program (Stewart et al., 1987; Stewart, 1991) that investigated the synoptic and mesoscale structure of Canadian East Coast winter storms, and field campaigns such as the meteorological monitoring network established for the Vancouver 2010 winter Olympics (Joe et al., 2014), and the Olympic Mountains experiment (OLYMPEX) that studied the modification of Pacific storms by coastal mountain ranges (Houze et al., 2017). Notably, the Vancouver Olympics network utilized a first-generation hotplate precipitation gauge, and while OLYMPEX used similar instrumentation to SAJESS such as disdrometers, weighing rain gauges, and micro rain radars, they also employed instrumented aircraft and a greater range of radar options. Ongoing research into East Coast snowstorm-producing cyclones is being undertaken by the Investigation of Microphysics and Precipitation for Atlantic Coast-Threatening Snowstorms (IMPACTS) campaign. Similar to OLYMPEX, IMPACTS combines surface observations and measurements with airborne remote sensing instruments. In terms of field projects, SAJESS compares closest to the recent (2019) Storms and Precipitation Across the continental Divide experiment (SPADE) (Thériault et al., 2021a, 2022), adopting the methods of manual observations, macrophotography, and micro rain radar and disdrometer deployment, although we built on SPADE by adding a distributed network of temperature and precipitation measurements, and upper air observations.

Locally, previous studies encompassing the Saint John River Basin have focused on flooding (Newton & Burrell, 2016), including rain-on-snow events (Buttle et al., 2016), and the analysis and modeling of ice jams that may increase in frequency in future climate scenarios (Beltaos et al., 2003). Despite these hazards, no studies of storms and precipitation and their impact on snowpack evolution have been conducted in this region.

The SAJESS dataset contains meteorological and precipitation data that were collected at a fixed station from 1 December 2020 until 30 April 2021, and an intensive observation period (IOP) that took place from 8 March to 30 April 2021. The objective of this paper is to describe the data collected during SAJESS, provide examples of the measurements and how they can be combined to broaden the picture of meteorological conditions observed, and to illustrate to stakeholders and potential users the importance of the field campaign and dataset.

The paper is organized as follows: the sites used during SAJESS are described in Section 2; the instruments, manual observations, and other sources of data are detailed in Section 3; an overview of data processing,

management, and validity, along with examples of data collected throughout SAJESS are presented in Section 4;
and a summary, including a discussion on potential future analyses, is given in Section 5.
**2 Site Descriptions**
**2.1 Overview**
To observe the spatial and temporal variability in precipitation amount and phase across the study area, a broad
range of techniques was employed: first, a semi-permanent 'Precipitation Phase Observatory' was installed to
record meteorological data from December 2020 to April 2021. This site became known as the 'Fixed Station'
and was co-located with the permanent ECCC station, north of Edmundston (Fig. 1). Second, the Mobile Urban
Station (MUST), a modified enclosed trailer provided by the Université du Québec à Montréal (UQAM), was
situated at the confluence of the Madawaska and Saint John rivers for the IOP during April and May 2021. The
MUST was located on property provided by the City of Edmundston and was used as a base for graduate students
and volunteers to record manual observations, capture macrophotography images of hydrometeors, and release
sounding balloons for upper air observations. Finally, community volunteers were engaged by providing locations
for either a satellite dish associated with the Smart Rainfall System (SRS) array (Coli et al., 2019), or a
precipitation gauge and snow board for the Community Collaborative Rain, Hail, and Snow Network
(CoCoRaHS) (Cifelli et al., 2005). Furthermore, 10 grade 6 classes (11-12 years old) from local elementary
schools also enrolled as CoCoRaHS observers.

**2.2 Precipitation Phase Observatory**
The Precipitation Phase Observatory (henceforth, the Fixed Station) encompassed a semi-permanent array of
meteorological instruments that were installed ~100 m from the Edmundston ECCC station on 30 November 2020
(Table 1, Figs. 1 and 2). The site was situated at the southern end of an area of open grassland in a broad valley,
152 m above sea level (ASL). The valley is 120-200 m wide by 1 km long, oriented north-south, and bordered by
coniferous forest. The 14-ha site acts as the municipal aquifer resupply and was provided by the city of
Edmundston for the installation. We provide specific location details in Table 1. This site was chosen to allow for
the SAJESS datasets to be supplemented and compared with records from the nearby ECCC station, which was
comprised of a Geonor T-200B weighing precipitation gauge with a single Alter shield, three sonic ranger snow
depth sensors, three Temperature/RH probes, and a RM Young wind monitor atop a 10-m mast. Additionally, the

 open field provided an opportunity to install an Infrared Gas Analyzer and Sonic anemometer (IRGASON) to

estimate surface turbulent fluxes and compute surface energy balances (Table 2) during the IOP (see Section 2.2).

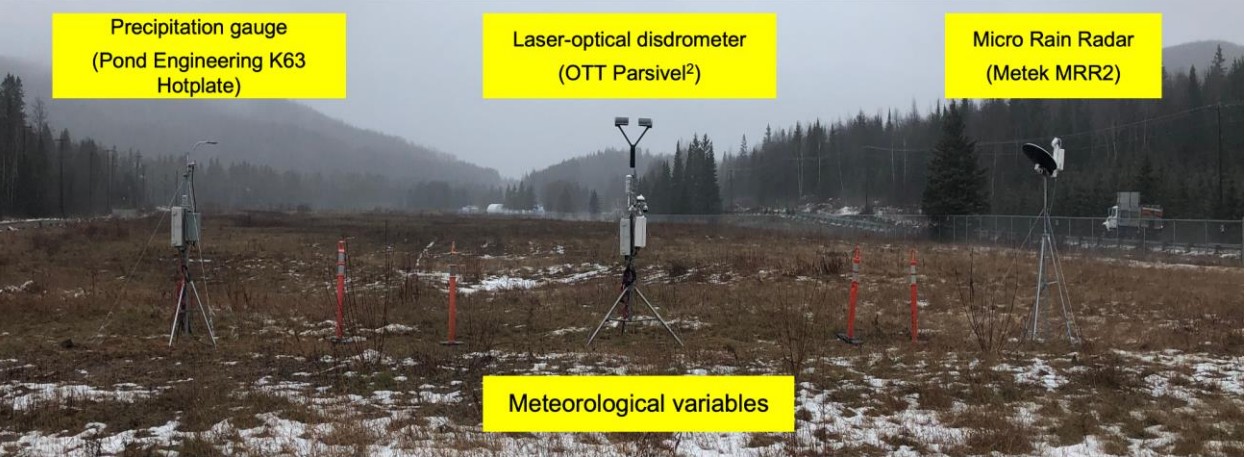


**Figure 2: The Precipitation Phase Observatory instrumentation, image taken looking north. From left-to-right: The K63 Hotplate, a laser-optical disdrometer installed upon the meteorological tripod, and the Micro Rain Radar. This station was also to be known as the 'Fixed Station'. Picture taken 1 December 2020.**


The identification of precipitation phase was achieved at the Fixed Station by the installation of a    laser-optical
disdrometer    for recording particle size and fall speed, and by a vertically pointing micro rain radar (MRR) to
provide information on the atmospheric conditions aloft (see Section 3 and Table 2). A K63 Hotplate Total
Precipitation Gauge (henceforth, hotplate) was installed to measure precipitation rate and amount. Aside from
periods of missing data (~5%), the Fixed Station dataset, excluding the Flux Tripod (see Section 3.1.2), spans 1
December 2020 – 30 April 2021.
**2.3 Intensive observation period**
Due to limitations at the Fixed Station (e.g., no fuel or generator use), a separate IOP site was established so that
the Mobile Urban Weather Station Trailer (MUST) and instruments could frequently be visited by observers (Fig.
1). The 6´ × 12´ enclosed trailer was equipped with heating, AC power, helium cylinders, and instrument storage;
it was parked on a fenced parcel of land on the north bank of the Saint John River and the east bank of the mouth
of the Madawaska River, 143 m ASL. Although the 3.3-ha site is dominated by the Edmundston wastewater
ponds, there was sufficient space for the MUST Trailer and instrumentation to be placed along the northern edge
of the site (Figure 4).

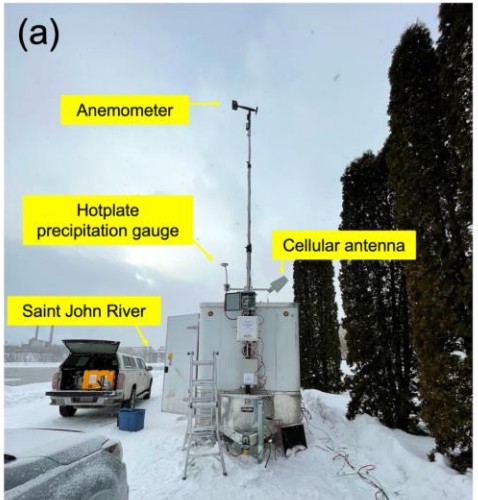
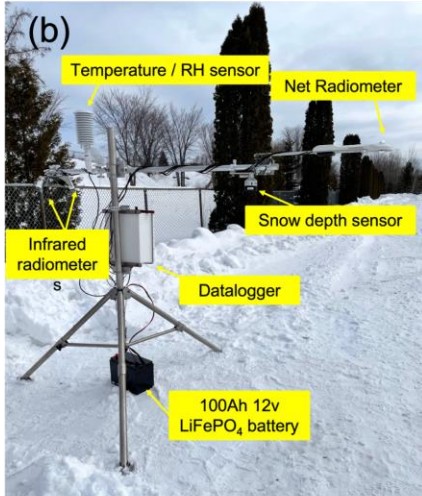
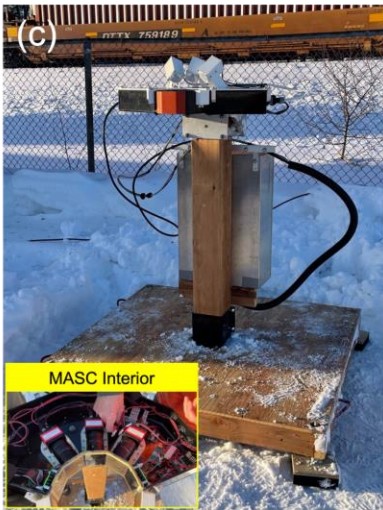
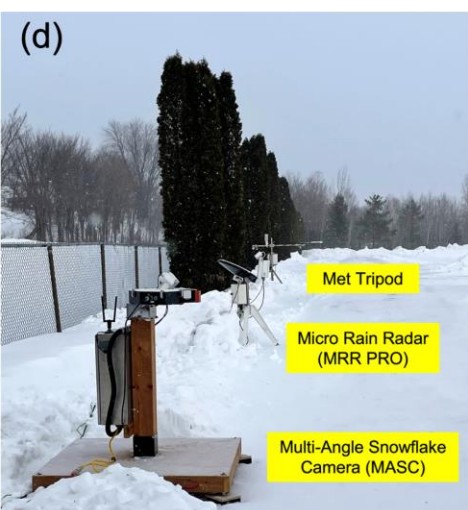


**Figure 3: Instruments and sensors co-located with the Mobile Urban Station (MUST) trailer. (a) The MUST trailer with extended 10 m mast, anemometer (not included in the dataset), and K63 Hotplate, (b) the meteorological tripod, (c) the Multi-Angle Snow Camera (MASC) with a top-down view of the internal components and three high-speed cameras, and (d) the MASC, MRR Pro, and meteorological tripod lined along the access road to the water treatment lagoon. Pictures taken 3 March 2021.**


The proximity of the instruments to the open ponds, nearby railway, and urban environment, resulted in the focus
on manual observations of weather conditions and precipitation type, rather than automated instrumentation, at
this location. We therefore stationed equipment that required regular attention or manual operation, such as the
multi-angle snowflake camera (MASC) (Figure 3c) and macrophotography equipment (not shown). Manual
observations and macrophotography (see Section 3.2) were conducted at the site from 1 March 2021 – 27 April
158 2021.

**2.3 Smart Rainfall System (SRS)**
The Smart Rainfall System (SRS) was installed at    six sites in the Edmundston area to capture the spatial
variability of precipitation by exploiting the satellite-to-earth links technology (Colli et al., 2019). Some locations
were also chosen to provide measurements upstream and downstream of Edmundston. The SRS system uses a
standard parabolic dish to receive satellite telecommunication broadcasting signals and an algorithm converts the
signal attenuation to precipitation rate (section 3.4). Locations with parabolic dishes that were already installed
by community volunteers for telecommunication purposes, but not being used during the experiment period, were
selected. Locations are shown Fig. 1.
**2.4 Community involvement (CoCoRaHS)**
While the CoCoRaHS network provides a broad array of precipitation measurements across North America
(Reges et al., 2016), there were few CoCoRaHS observer sites in the SAJESS study region. With assistance from
CoCoRaHS Canada (Colorado Climate Center, 2017), SAJESS students and staff facilitated the distribution of
equipment and training to a total of 21 new CoCoRaHS stations (Table 4), including 10 elementary schools, during
the 2020-2021 winter season. CoCoRaHS site metadata available to the public can be found at
https://cocorahs.org/Stations/ListStations.aspx.
**3. Field instruments and manual observations**
**3.1 Instrumentation**
The main consideration when deploying instruments to SAJESS was how to measure the amount, phase, and type
of precipitation that occurs during the winter and spring seasons. Particular attention was paid to gathering data
throughout as much of the tropospheric column as possible.

Here we provide an overview of the instrumentation used during SAJESS. Sensor details, parameters, and units
are included in Tables 2 and 3. Further details of each sensor, such as the date of manufacture, last calibration,
and serial number, are also provided in the readme files.

### 3.1.1 Meteorological tripod

Standard meteorological variables were measured at each of the two main sites. Common parameters for each
site were 2-m air temperature and relative humidity, 4-way net radiation (upwelling and downwelling long-wave
and short-wave radiation, LW↑, LW↓, SW↑, and SW↓), surface temperature, and snow depth. Measurements
at the Fixed Station also included soil surface temperature and moisture content. Snow at the Fixed Station
remained undisturbed for the winter and total snow depth was recorded continuously for December 2020 –
April 2021. Due to disturbance of the snowpack surrounding the MUST Trailer by foot traffic, vehicles, and the
installation of other instruments, a 30 cm × 30 cm white snow board was placed underneath the snow depth sensor
and cleared after each significant snowfall event. The clearing of the snow boards was consistently performed by
the volunteers/students. They were cleared at the beginning of observations before precipitation started, then
cleared again once precipitation ended. Through these actions, the ensuing data should be considered by users as
a snowfall measurement rather than true snow depth. On each met tripod, the snow depth sensor was installed on
the south end of the tripod cross arm so that the legs of the tripod would not be within ~1 m diameter cone of
detection. For surface temperature, two infrared radiometers (IRRs, henceforth IRR 1 and IRR 2) were mounted
to the north end of the cross arm of each met tripod and angled away ~ 30 degrees. At the Fixed Station, IR 1
faced slightly west, while IR 2 faced slightly east. At the MUST Trailer site these were reversed. The net
radiometer and snow depth sensors were mounted on the south end of each cross arms.

### 3.1.2 Flux tripod

An Open-path Eddy Covariance system was installed at the Fixed Station for the IOP from 5 March 2021 to 30
April 2021. The integrated Infrared Gas analyzer and Sonic Anemometer (IRGASON), temperature/RH probe,
net radiometer, infrared radiometer, soil probes, and heat flux plates were installed following the prescribed
methods found in Campbell Scientific (2022a), with the IRGASON sensor facing north into the prevailing wind.
Winds (in 3-D), air temperature, ambient pressure, and $CO_2$ and $H_2O$ densities were captured at 10 Hz resolution
and averaged over 30 minutes to calculate turbulent fluxes and energy closure balances (Campbell Scientific,
2022b). The dataset also includes diagnostic data, data quality values, and coefficients used for the eddy-
covariance calculations during each 30-min period so the raw time series data can also be post-processed using a
variety of software (US Department of Energy, 2022).

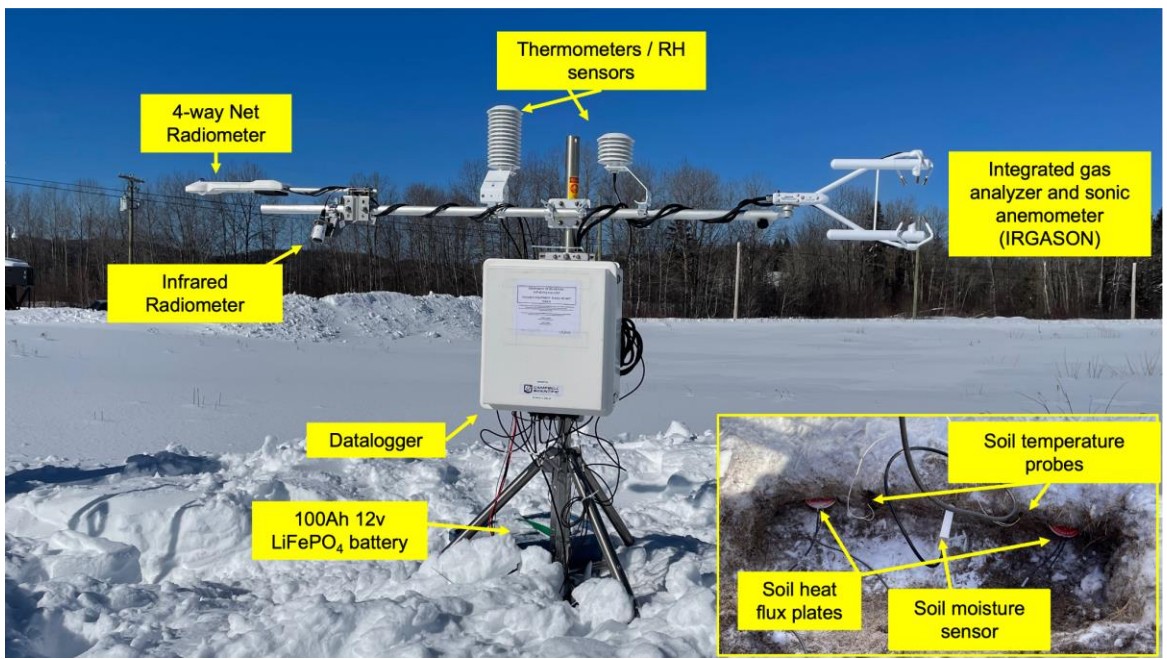


**Figure 4: The Fixed Station flux tripod. An open-path eddy-covariance system consisting of an Infrared Gas Analyser and Sonic Anemometer (IRGASON), soil temperature probes, a soil moisture sensor, a net radiometer, an infrared radiometer, and temperature/RH sensors. This tripod was installed for the melt period from 5 March to 30 April 2021. Picture taken 5 March 2021.**

### 3.1.3 Hotplate precipitation gauge

A hotplate precipitation gauge (henceforth, hotplate) was installed at the eastern end of the Fixed Station instrument array (Fig. 3) (Rasmussen et al., 2011; Thériault et al., 2021b). As outlined by Cauteruccio et al. (2021), the hotplate was tested during the World Meteorological Organization's Solid Precipitation Intercomparison Experiment (SPICE) (Nitu et al. 2018), and the Global Precipitation Measurement Cold Season Precipitation Experiment (GCPEX) (Skofronick-Jackson et al. 2015). The hotplate measures liquid-equivalent precipitation by recording 1-minute and 5-minute running averages of precipitation rate and wind speed (needed as control for precipitation rate). An accompanying environmental sensor measures air temperature, RH, and atmospheric pressure with the same 1-minute and 5-minute temporal resolution. SAJESS was the first time in Canada that the

Pond Engineering version of the hotplate was used in a field campaign. A second one of this type of hotplate was
installed for the IOP on the MUST Trailer mast and is detailed in Section 3.1.7.

### 3.1.4 Disdrometer

A laser-optical disdrometer was deployed at the Fixed Station for the duration of the field campaign on the same
tripod as the standard meteorological instruments, at 2.8 m AGL (Fig. 3). The disdrometer provides measurements
of the size and speed of falling hydrometeors which, when post-processed accordingly, can provide a classification
of hydrometeor type (Hauser et al., 1984; Rasmussen et al., 1999; Löffler-Mang and Joss, 2000; Ishizaka et al.,
2013; Thériault et al., 2021a). Included in the dataset are variables derived by the manufacturer's software such as
precipitation intensity (mm h-1), number of detected particles, and several national standard present weather
codes. The spectrum of particles is provided by 1024 columns representing the $32 \times 32$ matrix of particle fall
speed and diameter bins. The mid-values and spread of these bins are listed in the readme file supplied with the
dataset and in the disdrometer user manual (OTT, 2019).

### 3.1.5 Micro Rain Radar

A micro rain radar (MRR) at each of the primary sites was used during SAJESS to vertically profile hydrometeor
reflectivity and Doppler velocity (Tokay et al., 2009; Souverijns et al., 2017). A METEK MRR-2 was installed at
the Fixed Station, 2.6 m AGL, at the western end of the instrument array (Fig. 3). The MRR-2 has 32 range gates
(height steps) and we set the height resolution to the maximum of 200 m, giving a maximum height of 6400 m.
Raw data from the MRR-2 were post-processed using the IMProToo algorithm from Mahn and Kollias (2012), as
outlined in Thériault et al. (2021a). An MRR-Pro was installed at 1.3 m AGL at the MUST Trailer (Fig. 4d).
Settings for the MRR-Pro were: 128 range gates, 64 spectral lines, 10 s sampling, 50 m height resolution, and
6350-m ceiling. The combination of these parameters results in a velocity range of 12 m s$^{-1}$, and a velocity
resolution of 0.19 m s$^{-1}$. The 10 s averaging time results in a total of 305 spectra being averaged for each
measurement. Both radar units used built-in dish heating to eliminate snow and ice build-up during precipitation
events.

### 3.1.6 Multi-Angle Snowflake Camera

A multi-angle snowflake camera (MASC) was installed at the MUST Trailer (Fig. 4c and d), on a wooden stand
~1.4 m above the ground, for the majority of the IOP (5 March 2021 – 27 April 2021). Accumulating snow around
the base of the stand was removed by students to prevent the re-suspension of previously captured particles (Fitch
et al., 2021; Schaer et al., 2020). The MASC consists of three high-speed cameras housed in a single enclosure,
with 36° separation between each camera, and a focal point ~10 cm for each lens. The cameras take images
simultaneously when particles are detected within a ring-shaped viewing area (Fig. 4c inset). The three images
from each trigger consequently show the hydrometeor(s) from slightly different angles. First introduced by
Garret et al. (2012), MASCs have been deployed in Antarctica (Praz et al., 2017), the Colorado Rockies (Hicks
& Notaros, 2019), and Alaska (Fitch et al., 2020). Images have been used individually to illustrate the
hydrometeors observed, and highlight the riming conditions, crystal habit, size of particles.
**3.1.7 MUST Trailer mast**
A telescopic pneumatic 10-m mast attached to the MUST trailer supported an anemometer and wind vane (at 10
m), a hotplate precipitation gauge at 3.5 m, and the antenna for upper air observations (see Section 3.3) (Figure
4a). Due to air leakage, the mast did not always maintain its full extension and therefore required re-extending at
times. This, in addition to the proximity of the trailer to trees and a nearby structure has resulted in us excluding
the wind data from the publicly available dataset.
**3.2 Macrophotography, manual observations, and timelapse images**
Observers were present at the MUST Trailer during periods of precipitation to report weather conditions, and to
obtain macrophotographs of solid hydrometeors (Gibson and Stewart, 2007; Joe et al., 2014; Thériault et al., 2018;
Lachapelle and Thériault, 2021a). This provided a running-record of the weather conditions during storm events
and allowed for the field-training of students in manual observations, and identification of precipitation types and
snow crystal habits. The recording of manual observations can also remove doubt about precipitation arriving at
the surface. This is especially important with respect to hydrometeor type during periods of near-freezing
conditions, during which a mixture or changes in precipitation phase or type can occur. This also reduces the
potential for misdiagnosis by instrumentation or modeling. Observations of sky condition, cloud type, and
precipitation type (solid, liquid, or mixed), and images from a digital single lens reflex (SLR) camera fitted with
a macro lens and ring flash, were taken every 10 min. Precipitation was collected on a black velvet collection pad
during a short period (5 - 30 seconds), depending on the precipitation rate. The aim was to ensure that there were
enough particles to represent the precipitation conditions, while avoiding particles overlapping each other. The
collection pad was wiped every time after the series of nine images were taken using a set sequence of movements
of the camera upon a sliding frame. Images of a ruler placed across the pad for scale were taken intermittently,
for example when the camera was removed/replaced on the stand. Users can assume the scale remains constant
between these images (each image is 2 cm across). The scale images are listed in the macrophotography readme
file. More details of the method, including images of the equipment, can be found in Thériault et al. (2021a) and
Thériault et al. (2018). Other conditions to note were the occurrence of very light precipitation which can be
missed by disdrometers, and the presence of blowing snow that could affect analysis of MASC data (Section
286  3.1.6).


Hourly images of each site, including the surface conditions around the instruments, were captured by a time lapse
camera. Similar previous campaigns by the authors have found time-lapse images to be very useful in confirming
sky and surface conditions such as snow-on-the-ground onset and they have also provided evidence of wildlife
encounters with the instrumentation. Precipitation type diagnoses, however, are not usually possible with these
images.

### 3.3 Upper air observations

Soundings were timed to coincide with the standard synoptic times of 00, 03, 06, 09, 12, 15, 18, or 21 UTC. Some
additional launches were attempted to coincide with precipitation phase transitions. Sonde retrieval was not
attempted during this experiment. A total of 52 balloons were launched from the MUST Trailer and we include
text files for 46 of those launches that resulted in a complete sounding of the troposphere from which profiles of
pressure, temperature, dew point and winds (speed and direction) can be produced. Files of each sounding are also
available from the authors in other commonly used sounding formats.

### 3.4 Smart Rainfall System

SAJESS served as the opportunity to deploy an innovative environmental monitoring technique, the Smart
Rainfall System (SRS), that has been developed by the University of Genoa, Genoa, Italy and currently distributed
by Artys srl. The SRS produces estimations of liquid precipitation, in 1-min rainfall intensity, by processing the
attenuation of the satellite microwave link (SML) signal emitted by commercial geosynchronous satellites for
Digital Video Broadcasting (DVB-S) and received by common parabolic antennas (Colli et al., 2019). Estimating
liquid precipitation using the SRS has been confirmed by several experimental initiatives (Giannetti et al., 2021).
In contrast, snowfall intensity retrieval at centimeter wavelengths (the DVB-S signal is transmitted in the Ku
frequency band) is more uncertain. It has been demonstrated that higher operating frequencies, and preferably
dual-band systems, are needed to successfully retrieve solid precipitation (Falconi et al., 2018; Liao et al., 2016).

SAJESS provided the experimental conditions for SRS devices to monitor the liquid content in cases of mixed
precipitation and wet (melting) snow. The SRS system tested in Edmundston was composed of a set of distributed
SML sensors, as described by Colli et al. (2019), connected to a central processing and analysis node to reconstruct
the bi-dimensional rainfall field in real time. To an onlooker, the only equipment visible is a small box placed
inside the residence that connects inline to the satellite dish coaxial cable. The processor inside the box is then
configured to connect to the local Wi-Fi network. To reduce signal noise, it is preferable that the satellite dish no
longer be in service. These results will be published once work is complete.

### 3.5 CoCoRaHS sites


### 3.5.1 CoCoRaHS gauges and snowboards


Volunteers from the community and local elementary schools contributed to SAJESS by recording meteorological
measurements for the CoCoRaHS network (Cifelli et al., 2005). This provided an opportunity for students to
engage in the data collection process, and to learn about the importance (and difficulties) of precipitation
determination. A typical CoCoRaHS station includes a manual precipitation gauge to measure liquid and solid
precipitation, and a $40\,\mathrm{cm} \times 40\,\mathrm{cm}$ white board for measuring snow depth. Daily measurements include the amount
of precipitation, depth of snowfall, and snow water equivalent (SWE). Weekly measurements consist of total snow
depth, and the total SWE. CoCoRaHS data can be found using the network's online database; station details are
provided in Table 4. In addition to the regular CoCoRaHS station equipment, some volunteers hosted dataloggers
to record air temperature and relative humidity (see below).

### 3.5.2 Temperature sensors


HOBO MX2301A data loggers were distributed to 13 community volunteers (Fig. 1, Table 4), and correct
installation of each sensor was confirmed by a SAJESS team member. Installed approximately 2 m above the
ground, the dataloggers measure air temperature and relative humidity every 5 minutes (Onset Computer
Corporation, 2022). These low-cost, robust sensors provided a broad (50-60 km) network to assess spatial
variability in near-surface temperature/RH, especially during the passage of fronts and the onset/cessation of
precipitation. Data were retrieved from the HOBO devices via a Bluetooth smartphone app that reduced the need
to handle the sensor.

**4 Data description**

**4.1 Data processing and management**

Here we provide a short summary of the data processing and archiving strategies. Full details on all data, including specifications of the instruments used, can be found in the readme files uploaded to the FRDR repository.

Firstly, all instrumentation, camera equipment, observer notes, and computers were set to UTC date and time. Instruments that produce relatively low-volume text-based data such as the disdrometer, meteorological tripods, hotplate precipitation gauge, and temperature sensors, have been processed by concatenating smaller files together to create monthly files. Missing timestamps have been added to ensure every file contains timesteps for each minute of the month. All missing data points have been filled with NANs and no interpolation of missing data points has been attempted for the data uploaded to the FRDR repository. For standard meteorological variables such as temperature, humidity, snow depth, and radiation measurements, values have been quality checked to ensure they fall within the operating range of each instrument, with values outside of these ranges being set to NAN.

Raw radar data (.raw files) from the Fixed Station radar have been processed into daily NetCDF (.nc) format using the algorithm detailed by Maahn & Kollias (2012). Both .raw and .nc files have been included in this dataset. Hourly data from the MUST Trailer radar (MRR Pro) have been archived as .nc files as these data are produced by the instruments embedded processor (METEK, 2017)

Photographic images have not been altered or cropped and are uploaded as .png files for the MASC and macrophotography, or .jpg files for the timelapse cameras. Manual observations recorded in spreadsheets have been archived as comma separated value (.csv) files. Upper air observations are saved as one file per sounding, in tabular-delimited files, indexed by UTC date and time at a temporal resolution of 1 second.

In most instances, files have been identified using a specific naming convention using abbreviations for the project (SAJESS, SJ), each site (Fixed Station, FS; MUST Trailer MT), and each instrument (see Tables 2 and 3). For example, data from the disdrometer at the Fixed Station for the month of January 2021 are contained in the file: SJ_FS_DIS_MAS_202112.txt. (The abbreviation MAS stands for master and is used by the field staff to identify data that have been assembled ready for upload to the repository). MASC images do not follow this naming

convention as the software used with the instrument provides a detailed filename with respect to the snowflake
number and timestamp of the image.
**4.2 Data validity**
While not exhaustive, we list below known issues and attempts to-date at validating data from the SAJESS
instruments and observations. To preserve the raw data recorded during SAJESS, no quality control related post-
processing has been performed. We encourage all users to perform quality control and post-processing according
to their needs. We invite users to contact us for further information as several projects using SAJESS data are
ongoing.
**4.2.1 Met Tripod**
To our knowledge the only instrument to suffer from a systematic error was the HMP155A temperature and
humidity probe located on the meteorological tripod at the Fixed Station. Incomplete grounding at the datalogger,
and the subsequent datalogger program, resulted in a bias of ~ -2.32°C when compared to temperature data from
the ECCC station (Fig. 5). Due to the uniform pattern of the bias resulting in a low RMSE, these data have been
retained in the dataset and are available for use. Correction of these data by +2.32°C in on-going snow modeling
analyses indicate these data are still useful if post-processed accordingly. Post-deployment testing indicates that
this bias is not present in the temperature data from the MUST Trailer met tripod.

While data from the hotplate precipitation gauge include air temperature and humidity that align well with the
ECCC station (i.e., do not require bias correction), these data have significantly more noise than data from the
meteorological tripod sensor.

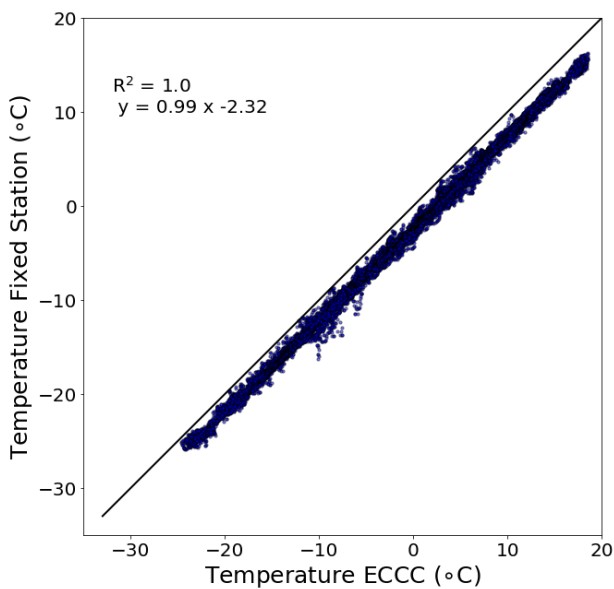

**388**

**389** **Figure 5: SAJESS Fixed Station and ECCC temperature data. Comparison of 1-minute temperature data (63149**

**390** **recordings) from the Fixed Station temperature probe and the mean of the three ECCC temperature thermistor**

**391** **readings. The Fixed Station HMP155 has a -2.32 °C bias due to the method of wiring and data recording.**

**392**

**393** Analysis of the IRR surface temperature data indicates that shading from the tripod and datalogger enclosure may

**394** have affected these measurements. When snow was present, the side with a shaded field of view (west in AM,

**395** east in PM) measured warmer conditions (up to ~2°C) than on the unshaded side. Once snow cover was absent,

**396** the trend was reversed so that the shaded side is cooler (up to ~6°C). This diurnal pattern of temperature difference

**397** between the two sensors existed at both SAJESS locations, so we caution against the averaging of both sensors

**398** without taking these differences into account.  We found similar results to Domine et al. (2021) where RMSE

**399** between the net-radiometer (using LW↑), and the IRRs was best reduced using an unphysical emissivity ($\varepsilon$) value

**400** of 1.028. This could indicate that direct comparison is difficult due to the difference in wavelength spectrum

**401** measured by the two instruments (Domine et al., 2021), yet the greater field of view of the net radiometer may

**402** also be influential. Surface temperature data have not been corrected for surface emissivity, and assume $\varepsilon = 1$

**403** (Apogee Instruments Inc., 2022). The location of the IRRs should be installed on the unshaded south end of the

**404** cross arm in future deployment.

**405**

Snow depth measurements from the sonic ranger (SR50A) on the Fixed Station meteorological tripod were within 10% of the values from the same type of sensor used by ECCC at the Edmundston station over the course of the 2020-21 winter. At both SAJESS sites, the 1-min resolution caused noise in the snow depth data during periods of precipitation, which is greatly reduced using a 1-hour running mean. Data from the SR50A also include a 'quality number', which is detailed in the dataset readme and can be used to filter snow depth data.

The 4-way net radiometers on each meteorological tripod were fitted with the heater/ventilation unit to reduce errors associated with dew/frost on the sensor window. This unit, however, does not heat the sensor window(s) sufficiently to remove snow or ice during or after precipitation, which is observed in the pyranometer data when SW↑ is greater than SW↓, and correlated with precipitation events using the disdrometer and/or snow depth data. This situation occurred on 13 days at the Fixed Station and five days at the MUST Trailer. For data analysis we suggest using more advanced algorithms such as Lapo et al. (2015) to identify periods of snow accumulation, which can then be corrected with other methods available in the literature (e.g., Sicart et al., 2006; Annandale et al., 2002). Unlike Domine et al. (2021), no sustained periods of 0.0 W m$^{-2}$ are observed in the LW↓ data, indicating the continuous use of the ventilation unit was successful in preventing frost build-up.

### 4.2.2 Flux tripod

Users wishing to utilize data from the flux tripod can investigate variables provided in the Flux Notes (FN), Campbell Scientific (CS), and AmeriFlux (AM) files. These include the number of $CO_2$, $H_2O$ samples per averaging period, and a corresponding 'bad data' column. We observe that the maximum number of samples at 10 Hz for the 30-minute period (18,000 in total) did not reach a total of 19.5 hours during the March-April 2021 IOP. These occurrences correspond to periods of precipitation whereby we presume the IRGASON windows were inhibited by rain/snow. On a finer scale, steady state integral turbulence characteristic (SSITC) tests were applied to energy balance components for every 30-minute period (Foken et al., 2004). These can be found in the AM files as variables named '_SSITC_TEST', and in the CS files as variables named '_QC'. Users should consult Foken et al. (2004) for further information regarding these tests as further analysis is beyond the scope of the paper. Data have not been gap-filled, removed, or replaced from the dataset.

(a)

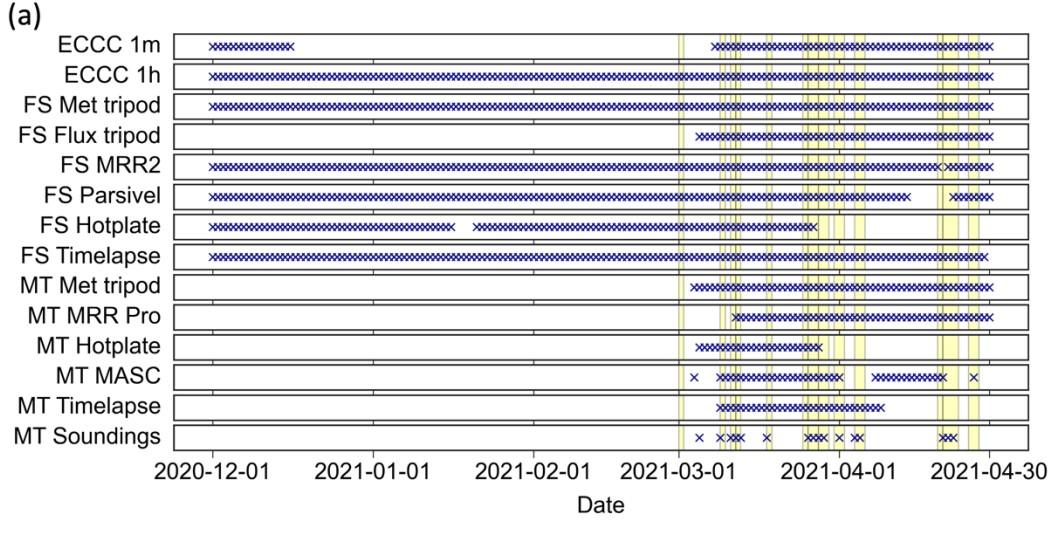

(b)

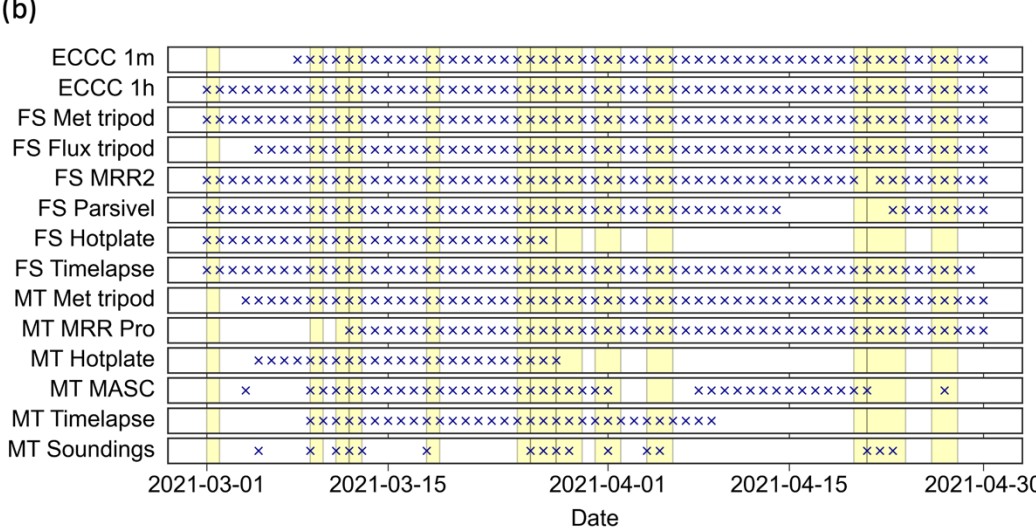


**Figure 6: Data availability (daily resolution). Operating periods for instrumentation for (a) the entire SAJESS field**
**campaign, 1 December 2021 – 30 April 2021, and (b) during the intensive observation period, 1 March 2021 – 30 April**
**2021 (bottom). Data availability for the Environment and Climate Change Canada station in Edmundston is given for**
**illustrative purposes only, as ECCC data are not included with this dataset. (FS stands for Fixed Station, and MT for**
**MUST Trailer). Vertical yellow bands indicate periods of manual observations at the MUST Trailer during storm**
**events by SAJESS volunteers and students.**

### 4.2.3 Hotplate

Although the hotplate performed well when fully operational, a significant portion of the IOP was missed (28 March 2021 to 30 April 2021) due to faulty microprocessor settings on both hotplates (Fig 6). These have subsequently been improved upon by the manufacturer, and further testing is underway. As described by Rasmussen et al. (2011), hotplate precipitation rate accuracy is least assured during the onset and cessation of precipitation, however, data from the hotplates include a 'Status' variable (#1, #2, or #3), that identifies these periods. A full explanation is provided with the dataset readme. Precipitation data from the hotplate have not been corrected for these under- or over-estimations.

Similar to Thériault et al. (2021b), we compared 30-minute cumulative sums of the hotplate 1-minute accumulation for Dec 2020 – March 2021, with corrected Geonor data (using Kochendorfer et al., 2017) from the Edmundston ECCC station. These align well with the best results (reduced bias and RMSE) found for rain (>2°C) and snow (<-2°C). Finally, hotplate T1 (1-minute average) precipitation data are more sensitive and therefore better at reproducing higher precipitation rates than the T5 (5-minute average) data, resulting in a positive bias up to ~1.5 mm h-1 (at 4-5 mm h-1).

Due to processor issues, barometer data were not recorded by the hotplate environmental sensor at all 1- or 5-minute intervals, however, valid readings do exist for each hour of the campaign. We recommend filtering barometer data from both SAJESS sites by selecting the median value during each hour to represent the hourly value. This method correlates well with ECCC station pressure reading (within 4 hPa) for the duration of the campaign. Barometer readings represent raw station pressure and are not corrected for elevation.

### 4.2.4 Disdrometer

The disdrometer dataset contains 1024 columns, which includes total particle counts and fallspeed, particle spectrum (Section 3.1.4) and other derived variables listed in Table 4. This configuration option is provided by the OTT software as raw data. At present we can only comment on a preliminary analysis of the OTT-derived precipitation intensity. When compared to 30-minute ECCC Geonor and hotplate precipitation rates, timing and amounts from the disdrometer are generally comparable, yet we observe the Parsivel[2] overestimates at high precipitation rates (>10 mm hr$^{-1}$), which is well documented (e.g., Angulo-Martínez et al., 2018). Users wishing to utilize the drop size and fall speed distributions, and subsequently retrieve an improved precipitation rate, can correct these data using Raupach and Berne (2015).

### 4.2.5 Upper air observations

Comparisons from four soundings from the MUST Trailer that aligned with either the 1200 UTC or 0000 UTC balloon releases from Caribou (ME) were made for data from 18, 27, and 29 March 2021, with all four sounding profiles displaying good agreement between the two sites. On some occasions, surface, and lower troposphere (<700 hPa) temperatures at Edmundston were up to 5°C cooler than in Caribou, ~60 km to the northwest. The SAJESS sounding data also correlate well with the surface observations of precipitation type and phase. We recommend smoothing the 1 s sounding data using a low pass filter (e.g., 10 s running mean) to remove noise in the temperature and dewpoint profiles.

### 4.2.6 CoCoRaHS precipitation reports

CoCoRaHS volunteers conducted most of their observations during the IOP, due to the delays in obtaining equipment and training due to public health measures in place at the time. Despite this, 21 CoCoRaHS stations (Table 4) reported 1715 daily total precipitation (liquid equivalent) measurements, 1729 new snow measurements, and 1123 total snow measurements (Fig. 7). A comparison of the daily precipitation and new snow amounts has not been conducted, but the total snow (i.e., snow on the ground) compares well with the Fixed Station and ECCC instruments. CoCoRaHS data are not included in the SAJESS dataset as: (a) CoCoRaHS already provide long-term storage and data retrieval via its own website, and (b) the authors encourage users to explore the CoCoRaHS database for stations that may be useful for their own analysis that were not associated with SAJESS.

### 4.2.7 MASC

The MASC captured a total of 93,343 during March-April 2021. We present some of these images in Section 4.3.3, where they corroborate the particle type recorded by the manual observations. The MASC also allowed for automated photography of particles when observers were not able to attend to manual observations or macrophotography. Users should expect, however, a large portion (perhaps 50%) of images to not contain particles distinguishable to the naked eye. We attribute this to SAJESS representing an experimental opportunity to deploy the MASC in mixed-phase precipitation conditions. While images of liquid or freezing precipitation were usually blurred and would be considered less than ideal from an aesthetic point of view, post-processing of the images using algorithms provided in Praz et al. (2017) has resulted in particle fall speed and size distributions (similar to disdrometer results) that align well with the observed precipitation type. Analysis of the images is ongoing.

## 4.3 Examples of data and observations

Here we provide an overview of the conditions observed during the SAJESS campaign, and examples of data from each of the two SAJESS sites to illustrate possible uses. While the total SAJESS dataset is ~200 Gb, a ~1.1 Gb sample of data is available on the FRDR repository. This sample dataset is based on the example given here for the MUST Trailer location, where a subset of data for 18 March 2021 is displayed. Sample data have been made available for most instruments for the entire day (0000 – 2359 UTC), and for the MASC, macrophotography images, and the MRRs for 1200 – 1300 UTC 18 March 2023 (to reduce file size).

### 4.3.1 Overview

Data covering 1 December 2020 – 30 April 2021 are provided by the instruments at the Fixed Station, which can be supplemented by the near-by ECCC station. Maximum snow depth measured at the Fixed Station was 65 cm, with snowfall amounts comparing well in timing, yet slightly less in magnitude (around 10%), than measured by the ECCC instruments (Figure 7). Maximum snow depth observed at 21 surrounding CoCoRaHS stations (Table 4) ranged from 50 cm to 79 cm, with an average of 54 cm. During this period, the ECCC Geonor measured a total of 287 mm of (liquid equivalent) precipitation, less than the climatology (1993-2022) mean of 364 mm.

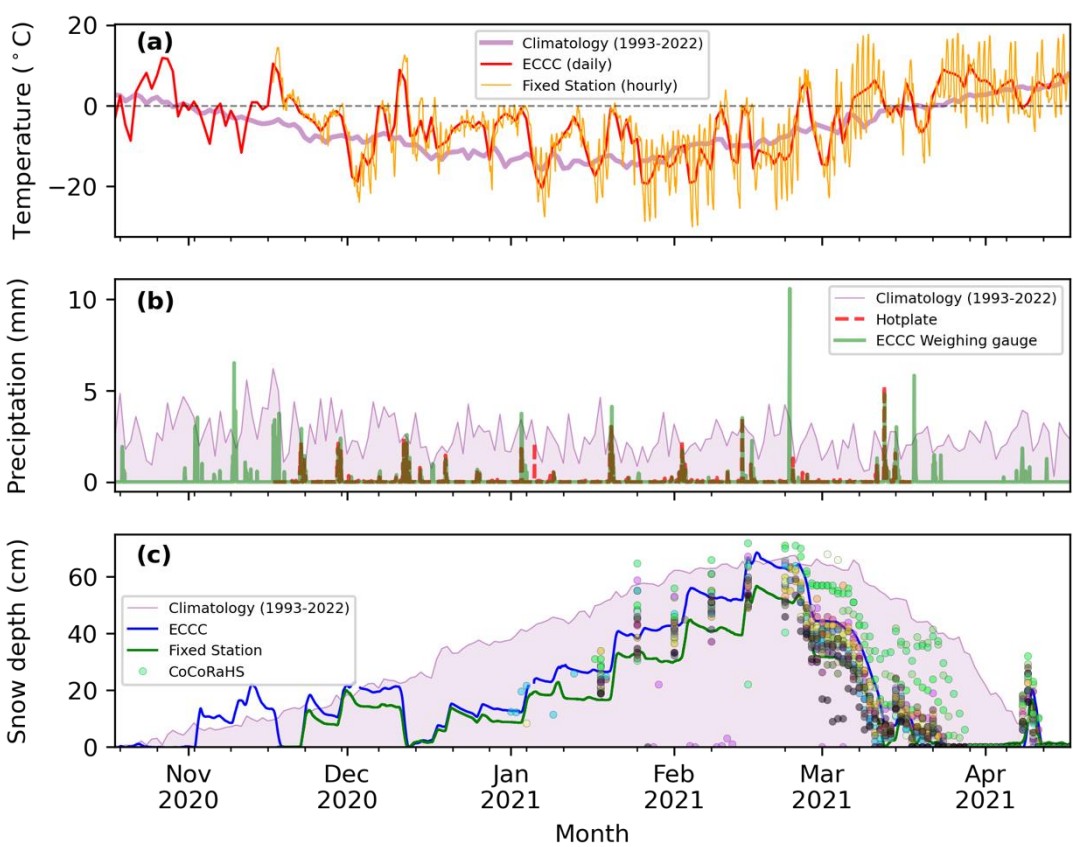

512

**Figure 7: Overview of select SAJESS measurements (December 2020 to April 2021) with comparison to the Edmundston ECCC station and 1993-2022 climatology. (a) Hourly mean Fixed Station temperature (corrected, see Section 4.2.1), daily mean ECCC temperature, and daily mean climatology temperature. (b) Hourly accumulation from the Fixed Station hotplate precipitation gauge and ECCC weighing gauge, with daily climatology accumulation. (c) Fixed Station and ECCC snow depth measurements, 1993-2022 snow-on-the-ground climatology, and measurements from local CoCoRaHS stations listed in Table 5 (each station has a randomly assigned individual marker color). No hotplate measurements are available, or therefore shown, for April 2021 due to an instrument fault.**

520

521 During the IOP (8 March 2021 - 30 April 2021) manual observations were made during 13 storms (Fig. 6), for a

522 total of 183 hours of precipitation-type observations. Observers were present for 93 hours of snow (8 storms), 63

523 hours of rain (5 storms), and 27 hours of mixed precipitation (more than one phase of precipitation occurring at

524 the same time; 3 storms). Two storms (12 March 2021; 4-5 April 2021) included rain-snow transitions. Observers

525 took a total of 4483 images of precipitation particles during the IOP. Using the ECCC Geonor, observers were

526 present for 97 mm of the 105 mm of (liquid equivalent) precipitation that fell during the IOP period.

### 4.3.2. Fixed Station storm measurements

Focused examples of the meteorological measurements at the fixed station are given Fig. 8. This case covers the period 25 – 27 December 2020 during which 22.5 hours of rain-on-snow (14 mm) from 1230 UTC 25 December to 1100 UTC 26 December 2020, with temperatures of 5°C – 6°C, reduced the Fixed Station snow depth to 0.0 m. This was followed closely by a decrease in temperature to < 0°C and ~4.5 hours of snow (4 cm) from 1630 UTC to 2050 UTC 26 December 2021), restarting the seasonal snowpack from bare conditions. Precipitation type derived from the disdrometer data correlates well with the decrease in air temperature below 0°C. The time series31 of particle type can be constructed from the binned fall speed and diameter measurements and by using equations in the literature such as Rasmussen et al. (1999) and Ishizaka et al. (2013), or by using the precipitation type codes produced by the instrument software (Table 2). MRR data show the reduction of melting height (determined by the sharp vertical gradient of reflectivity) from ~ 3 km AGL to just over ~ 2 km AGL during this same period. These insights to the evolution of the seasonal snowpack are rare as many standard weather or climate stations may not have the ability to explicitly identify precipitation types.

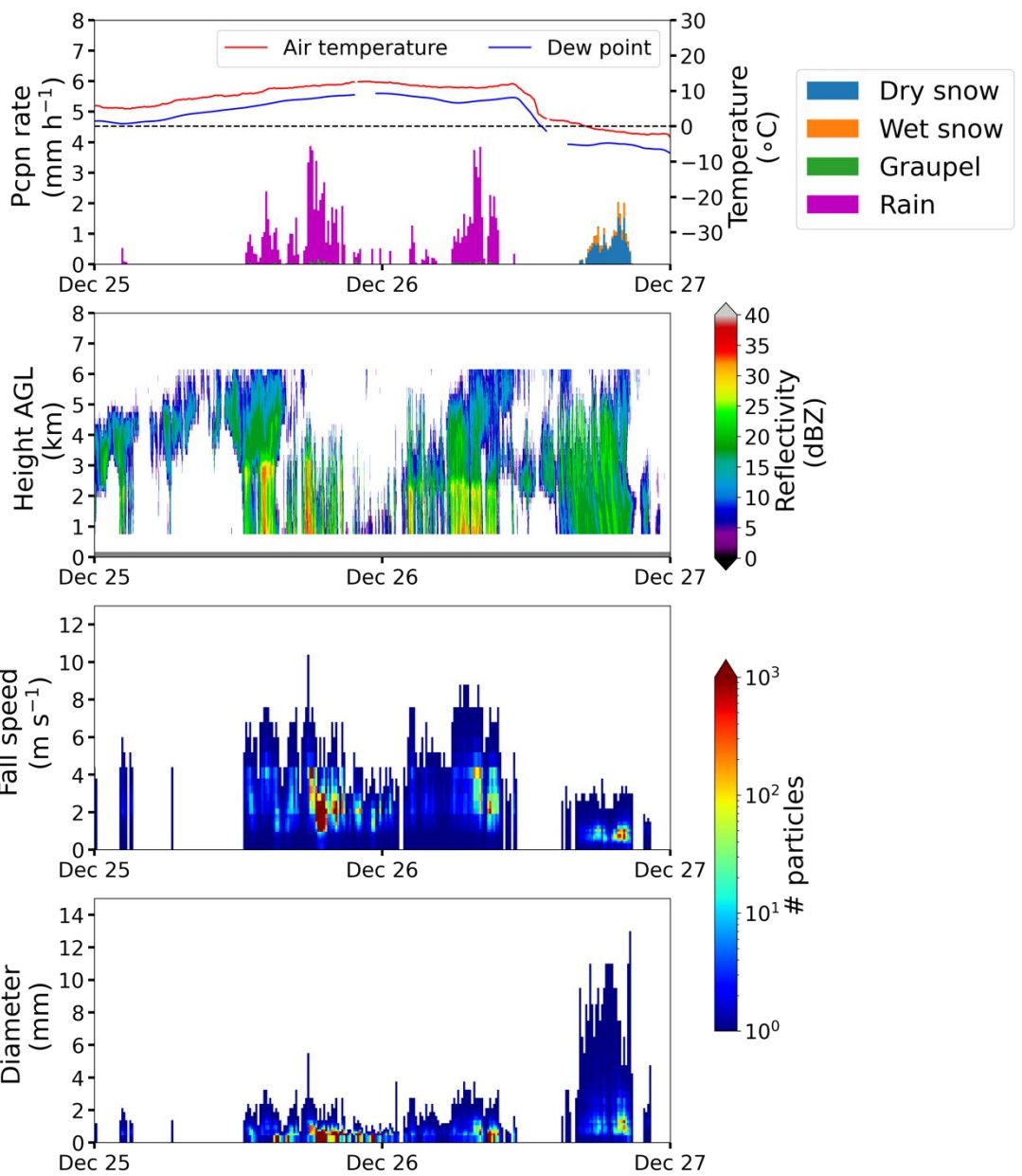


**Figure 8: Fixed Station storm measurements. From top to bottom, (a) air temperature and precipitation. The**
**precipitation type is derived from disdrometer data using equations in Rasmussen et al. 1999 and Ishizaka et al. 2013,**
**and is not included in the dataset. (b) radar reflectivity, (c) particle fallspeed, and (d) diameter (both from the Parsivel**
**disdrometer), measured at the Fixed Station from 0000 UTC 25 December 2020 to 2359 UTC 26 December 2020.**
**Temperature, dew point, precipitation, and disdrometer data are shown as 10-minute averages. Micro rain radar**
**(MRR) data are at 10 second resolution, with a vertical resolution of 200 m.**

### 4.3.3 MUST Station storm observations

To complement the automated measurements made at the Fixed Station, the MUST Trailer site provided observations of precipitation type, photographic imagery of particles, and upper air soundings, in addition to the meteorological tripod and hotplate. Images and observations taken during a snow event that occurred on 18 March 2021 are shown in Figs. 9 and 10.

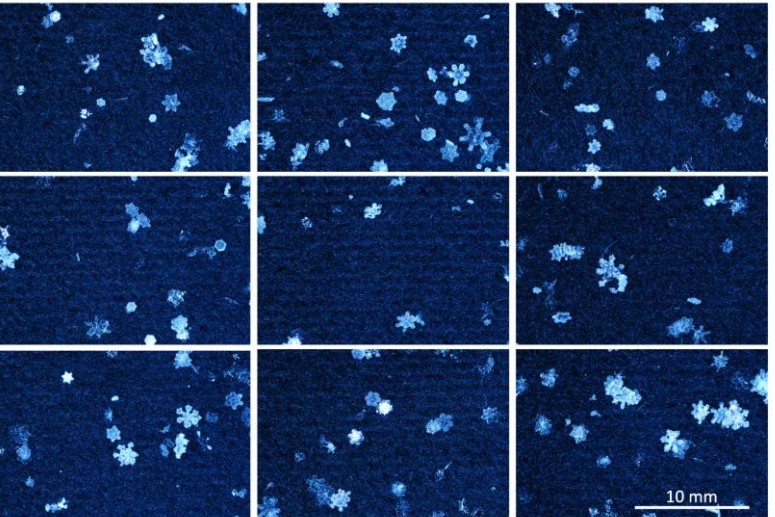

**Figure 9. MUST Trailer macrophotography. Nine independent (i.e., no overlap) images of the felt-covered pad with solid hydrometeors collected at 1225 UTC 18 March 2021. Images are oriented as they were on the 13 cm × 13 cm pad.**

From 1000 UTC to 1430 UTC 18 March 2021, light intermittent snowfall with precipitation rates between 0.4 – 0.6 mm h$^{-1}$ resulted in ~1.5 cm of snowfall at the MUST Trailer. Precipitation rates were at times so low that the hotplate did not record any precipitation, but snowfall was confirmed by the manual observations. The Fixed Station disdrometer also detected very light precipitation (maximum rate of 0.6 mm h$^{-1}$) with only 0.2 mm total (liquid equivalent) for the same period. The ECCC Geonor did not record any precipitation during this time. Air temperature at the MUST Trailer remained between -1°C and 0°C during most of the snowfall, with a short (~30 min) period of snow grains being reported when temperatures increased to 0°C – 0.5°C. Macrophotography images taken from the SAJESS MUST Trailer provide confirmation of hydrometeor type. From these images (Fig. 9) crystal habit, size distribution, and riming can be diagnosed, and they align with the 10-minute

observations that recorded overcast skies, nimbostratus clouds, and snow. Also, during this time, the MASC
captured images of a variety of hydrometeors, with the clearest images being of large aggregates (Fig. 10a). At
1200 UTC 18 March 2021 a deep saturated layer between 900 and 700 hPa is evident on the upper air observations
from the MUST Trailer, matching the vertical profile observed in the National Weather Service (NWS) sounding
from Caribou (ME) (Fig. 10b).

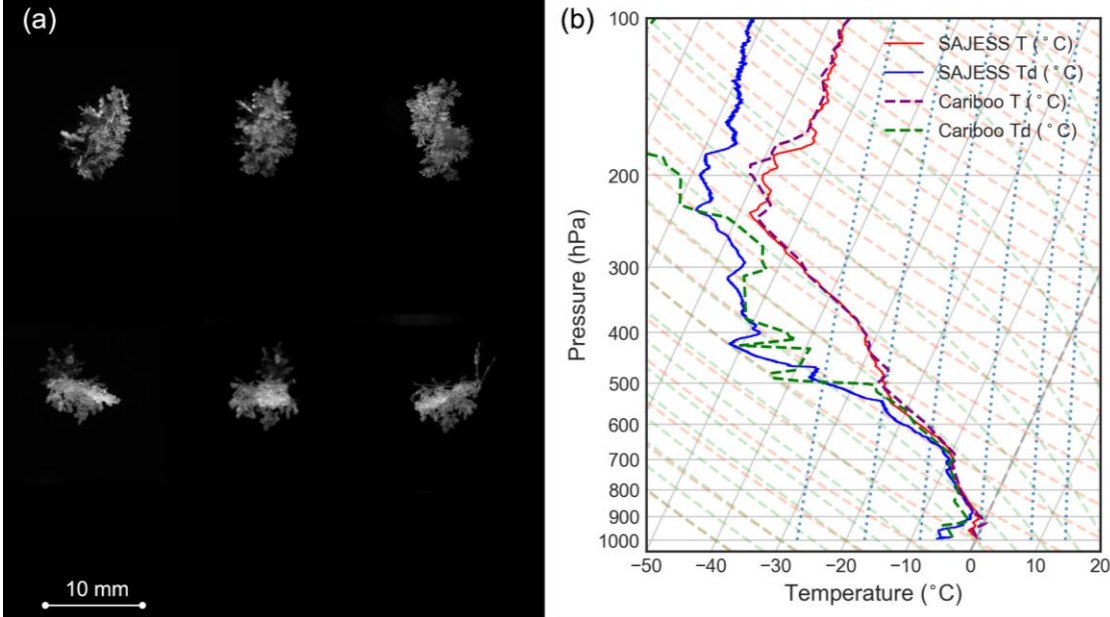

**Figure 10. Multi-angle snowflake camera (MASC) images and upper air observations. (a) Two triplet images of**
**aggregates taken by the MASC at 1240 UTC 18 March 2022, and (b) the 1200 UTC 18 March 2021 soundings from the**
**SAJESS MUST Trailer and the NWS Caribou (ME) station.**

After a break in conditions, light precipitation made up of wet snow and rain occurred between 1730 UTC – 1950
UTC 18 March 2021. The mixed precipitation erased the snow collected on the snowboard under the MUST
Trailer SR50A. This is important as it illustrates what CoCoRaHS observers would have encountered the next
morning (19 March 2021), when no observations of new snow were recorded.

## 4.4 Challenges and lessons-learned

Challenges during the SAJESS field campaign primarily originated from the requirement for remote computer access to monitor and troubleshoot instrument problems as they arose, and intermittent interruptions to AC power, especially at the Fixed Station site.

Such issues include the requirement to restart proprietary software. For example, the disdrometer software provided by OTT requires a user to restart the program manually if the computer has suffered a power interruption or reboot. This caused some periods of missing data to be longer than necessary. For subsequent deployments we have circumvented this issue by writing a serial-based terminal program in Python that can run on any operating system. Conversely, software provided by METEK for the MRR will restart automatically, however, users are limited to Windows operating software.

The processor included with the MRR Pro does allow for data collection to begin automatically, for as long as the on-board hard drive has space, however, raw data then are not retrievable as they are with the MRR2. We found this to be an adequate trade-off as the longer acquisition time per spectrum results in the MRR Pro signal having less noise, and due to the larger number of range gates, can have a finer vertical resolution for the same maximum height as the MRR2. Our use of both versions of the METEK MRR was also due to equipment limitations within the group.

A similar restart issue arose for the hotplate, whereby a manual reset was required after the instrument's microprocessor cut power to the heating plates. This response has been rectified by the manufacturer; however, we have since installed a remotely controlled AC outlet to our other hotplate stations that can be switched on/off without having to visit the site in-person.

SAJESS provided an opportunity to operate the multi-angle snowflake camera (MASC) during mixed-phased precipitation. Initial results indicate that the instrument can help diagnose mixed-phase precipitation using post-processing algorithms that diagnose particle fallspeed and diameter to a similar standard as the disdrometer. This leads, however, to many of the images being of blurred spherical raindrops rather than sharp pictures of ice crystal types published elsewhere. Work is ongoing to modify the post-processing software to include the categorization of raindrops, as this category is currently missing from the software.

Dishes for the SRS are required to be oriented to specific satellites (with the help of a satellite signal finder);
however, it is also best if these dishes are not currently being used to receive a satellite TV signal by the household.
In Edmundston, we located an adequate number of unused dishes within the community, however, an SRS specific
deployment may require the purchasing of new equipment solely for the purpose of the SRS system. In some
regions, satellite dishes may no longer be installed or maintained to a sufficient standard.

The use of community-based volunteers to assist in observations at times proved challenging, practically with
respect to data quality. Our conclusion is that SAJESS could have benefited from a smaller number of high-quality
measurements that could be verified, rather than a larger number of measurements that may have been prone to
error. Due to public health measures in place much of the hands-on training that was planned had to be conducted
remotely, and at times restrictions limited the volunteer's access to sites and equipment. We suggest a dedicated
team member to act as liaison for larger groups of volunteers. Compared to the high-temporal resolution of many
of the SAJESS instruments, precipitation amounts recorded once-per-day may not be suitable for all analyses, as
identified in Section 4.3.3.

Particularly difficult was the installation of the eddy covariance flux tripod during the winter season. The burial
of temperature sensors and heat flux plates was less than ideal in the frozen surface material, which also precluded
a soil pit analysis and soil layer documentation. However, care was taken to ensure restoration of the soil and turf
was as complete as possible after installation. The overlying snow was also restored as homogeneously as possible
to match the surrounding snowpack. Due to the lack of soil analysis, the datalogger program's default bulk density
value of 1300 kg m$^{-3}$ was used. Although future deployments should not underestimate the time and energy
required to install the flux instrumentation correctly, we have retained the flux data as thorough post-processing
has allowed for effective analysis of the data (Leroux et al, 2023).
**5 Summary**
A valuable dataset was collected during the 2020-2021 Saint John River Experiment on Cold Season Storms
(SAJESS) over Eastern Canada. The dataset contains automatic measurements, manual observations, and photos
of the meteorological conditions and precipitation at the surface, micro rain radar measurements, upper air
observations, and measurements of precipitation amounts by community volunteers.

The experiment included an intensive observation period during March – April 2021, to document conditions
during the melt season with its below-average snowpack melt a month earlier than normal. The study region and
downstream communities are historically prone to ice jams and flooding during late winter and early spring, which
can be influenced by air temperature, rain-on-snow events, and the overall evolution of the seasonal snowpack.
Despite these hazards being minimal during the campaign, the dataset will contribute to an understanding of
snowpack evolution across the region by a variety of mechanisms. Firstly, the spatially distributed measurements
from CoCoRaHS observers and SRS installations give a broader view of precipitation types and amounts than the
typical isolated weather or climate station. The dataset contains more than 1700 measurements from these
community volunteers, and the experiment doubled as a learning opportunity for the classes of school children
that contributed by recording snow board and rain gauge measurements. Secondly, the identification of
precipitation type at the surface, particularly when there is snow on the ground and temperatures are near 0°C,
can be vital for forecasting the resulting snowpack and hydrological response. This was achieved during SAJESS
by combining high temporal resolution measurements (1-minute) of precipitation amount, snow depth, and
disdrometer data, with manual observations and photographic imagery of hydrometeors at the surface. Verifying
the precipitation type using these measurements removes the uncertainty of precipitation type estimation that is
necessary without such an array of measurements. Thirdly, observations of atmospheric conditions and
precipitation aloft from the micro rain radar and upper air soundings will advance our knowledge of pre-storm
conditions and the influence of tropospheric conditions on the precipitation type observed at the surface. When
these conditions result in solid precipitation at the surface, inspection of the macrophotographs and MASC images
can be included to confirm particle type, size, and shape, and for snow, crystal habit and riming condition. The
SAJESS dataset provides an opportunity to investigate the micro-scale influences these differing particle types
and crystal habits have on snowpack evolution.

In addition, the SAJESS dataset provides the opportunity to advance forecasting and model evaluation. The
breadth of instruments and human observations quantifies many parameters not archived in traditional
meteorological databases. Examples of this include the non-standard timing of balloon launches, spatially
distributed precipitation and temperature observations, as well as energy balance and flux data. The identification
of particle type obtained by using non-standard observations such as the MASC provides ground truth to help
further model microphysics.

Finally, the SAJESS field campaign also highlights the need to enhance measurements of precipitation and snow
in the upper Saint John River Basin. Near real-time access to data during the campaign allowed for the monitoring
of meteorological conditions as they occurred, which can be vital for emergency management, ice jam and flood
forecasting, and river navigation. This is not usually available in areas where weather or climate stations are
sparse. We envision that consideration be given to furthering the meteorological monitoring network in the upper
Saint John River Basin to increase the resolution and frequency of measurements and to better anticipate ice jams
and major flooding events along the Saint John River.

**Author contributions**
HDT wrote the first draft of the manuscript, as well as conducted some analyses. JMT, SJD, RES, and VV
designed and led the field project. DB and LR collected manual observations during the intensive observational
period. NDL contributed to the installation of the instruments and the management of the CoCoRaHS observers.
MC provided the Smart Rainfall System (SRS). HDT, JMT, SJD, RES, DB, LR, NRL, MC and VV contributed
to the writing and the editing of the manuscript.

**Competing interests**
The authors declare that they have no conflict of interest.

**Data availability**
The SAJESS dataset (including the sample subset of data) is available from the Federated Research Data
Repository (FRDR) and can be accessed at https://doi.org/10.20383/103.0591 (Thompson et al., 2023), and is
included in the Global Water Futures FRDR collection. CoCoRaHS data are available from
https://cocorahs.org/ViewData/. SRS data are available from the Artys' web platform (https:/www1.artys.it/) that
can be accessed upon request (m.colli@artys.it). Hourly and daily data from the Edmundston ECCC station are
available via the ECCC Historical data webpage, with 1-minute raw data from the station available from the
authors upon request. (https://climate.weather.gc.ca/historical_data/search_historic_data_e.html).

**Acknowledgments**
Funding was provided by the Global Water Futures programme which is project 418474-1234 funded by the
Canada First Research Excellence Fund, Natural Sciences and Engineering Research Council of Canada
Discovery Grants (Julie M. Thériault, Stephen J. Déry, and Ronald E. Stewart), the Canada Research Chairs
Program (Julie M. Thériault), and UNBC (Lisa Rickard), to conduct scientific analysis. The MUST Trailer was
developed with funding from the Canadian Foundation for Innovation. Many thanks to all the volunteers and
schools who collected measurements and provided locations for the SRS parabolic dishes across northwest New
Brunswick during SAJESS. Thank you to Jacques Doiron, Director of the Emergency Measures for the City of
Edmundston, for providing the sites, facilities and coordinating the local activity with the SAJESS team, and
Amanda Ronnquist for creating the SAJESS data management plan.  We appreciate the detailed reviews and
constructive comments provided by Siwei He, Robert Hellstrom, and two anonymous referees, and assistance of
editor David Carlson.

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

**Table 1. Primary site information**

| Location | Lat (°N) | Lon (°W) | Elevation (m) | Surface | Surroundings | Dates of operation for SAJESS |
|----------|----------|----------|---------------|---------|--------------|-------------------------------|
| Fixed Station | 47.418 | 68.324 | 152 | Grassland on gravel riverbed | Open grassland in broad river valley, rural road ~150 m to the west | 1 December 2020 – 30 April 2021 |
| MUST Trailer | 47.361 | 68.320 | 143 | Packed gravel, short grass | Site on edge of city treatment ponds, ~ 250 m from confluence of two large rivers; suburban subdivision to the north, | 1 March 2021 – 30 April 2021 |


**Table 2. Instrument details for the SAJESS Fixed Station site**

| Installation (instrument abbr.) | Instrument | Installation height | Variable | Units | Resolution | Accuracy |
|---|---|---|---|---|---|---|
| Met tripod (MET) | Campbell Scientific CR1000X datalogger | 1.5 m | data recording | NA | NA | NA |
| | OTT Parsivel[2] disdrometer | 2.80 m | Timestamp<br>Intensity of precipitation (mm/h)<br>Precipitation since start (mm)<br>Weather code SYNOP WaWa<br>Weather code METAR/SPECI<br>Weather code NWS<br>Radar reflectivity (dBz)<br>MOR Visibility (m)<br>Signal amplitude of Laserband<br>Number of detected particles<br>Temperature in sensor (°C)<br>Heating current (A)<br>Sensor voltage (V)<br>Kinetic Energy<br>Snow intensity (mm/h)<br>V0D0 … V31D31 | Timestamp<br>mm h$^{-1}$<br>Mm<br>val<br>val<br>val<br>dBz<br>m<br>val<br>val<br>°C<br>A<br>V<br>J (m$^2$ h)$^{-1}$<br>mm h$^{-1}$<br>val | 1 min (average), except for the VxDx size and fallspeed classes, which represent a 1 min sum | Accuracy is not given by OTT for the individual variables, however the size classification are not given by OTT±1 size class (0.2 to 2 mm)<br>±0.5 size class (>2mm) |
| | Vaisala HMP155 temperature /RH probe | 2.00 m | Air temperature<br>Relative Humidity | °C<br>% | 1 min (average) | 0.226+0.0028×reading (−80 to +20 °C)<br>0.055+0.0057× reading (+20 to +60 °C) |
| | Kipp & Zonen CNR4 | 1.80 m | 4-way net radiation | W m$^{-2}$ | 1 min (average) | < 5 % |

| | Net radiometer | | | | | |
|---|---|---|---|---|---|---|
| | Campbell Scientific SR50A sonic ranger | 1.80 m | Snow depth | m | 1 min (average) | ±1 cm |
| | Apogee SI-411 IR radiometer | 1.80 m | Surface temperature | °C | 1 min (average) | ±0.2 °C |
| | Campbell Scientific CS655 Soil probe (vertically) | 0.00 m | Soil temperature<br>Soil moisture content<br>Soil electrical conductivity | °C<br>%<br>dS m$^{-1}$ | 1 min (average) | ±0.1 - 0.5 °C<br>±1 - 3%<br>±5% of reading + 0.05 dS m$^{-1}$ |
| MRR tripod (MRR) | METEK MRR-2 | 2.60 m | Doppler raw spectra<br>Reflectivity (Ze)<br>Doppler velocity (W)<br>Spectral width (σ) | dB<br>dBz<br>m s$^{-1}$<br>m s$^{-1}$ | 10 sec raw data; 1 min average; see Section 3.1.5 | 0.53 dB<br>0.53 dBZ<br>0.109 ms$^{-1}$<br>0.09 ms$^{-1}$ |
| Hotplate tripod (HP) | Pond engineering K63 Hotplate precipitation gauge | 2.60 m | Air temperature<br>Barometric pressure<br>Precipitation rate<br>Accumulation<br>Windspeed<br>Hotplate power | °C<br>kPa<br>mm hr$^{-1}$<br>mm<br>m s$^{-1}$<br>W | 1 min (average)<br>5 min (average) | ±1 °C<br>±1 kPa<br>±0.5 mm hr$^{-1}$<br>±0.5 mm<br>±1 ms$^{-1}$<br>NA |
| Flux tripod (FLUX) | Campbell Scientific CR1000X datalogger | 1.5 m | data recording | NA | NA | NA |
| | Vaisala HMP155 Temperature /RH probe | 2.00 m | Air temperature<br>Relative humidity | °C<br>% | 30 min (average) | 0.226+0.0028×reading (−80 to +20 °C)<br>0.055+0.0057×reading (+20 to +60 °C) |

| | Apogee SI-411 Infrared radiometer | 1.80 m | Surface temperature | °C | 30 min (average) | ±0.2 °C |
|---|---|---|---|---|---|---|
| | Kipp & Zonen CNR4 Net radiometer | 1.80 m | 4-way net radiation | W m$^{-2}$ | 30 min (average) | < 5 % |
| | Campbell Scientific IRGASON | 2.00 m | 3D wind<br>$CO_2$ density<br>$H_2O$ density<br>Sonic temperature | m s$^{-1}$<br>mg·m$^{-3}$<br>g·m$^{-3}$<br>°C | 10 Hz | 1 mm s-1<br>0.2 mg·m$^{-3}$<br>(0.15 µmol·mol$^{-1}$)<br>0.00350 g·m$^{-3}$<br>(0.006 mmol·mol$^{-1}$)<br>0.025 °C |
| | Campbell Scientific CS655 Soil probe | 2.5 cm below surface | Soil temperature<br>Soil moisture content<br>Soil electrical conductivity | °C<br>%<br>dS m$^{-1}$ | 30 min (average) | ±0.1 - 0.5 °C<br>±1 - 3%<br>±5% of reading + 0.05 dS m$^{-1}$ |
| | Campbell Scientific HFP01 Soil heat flux plates | 8 cm below surface | Soil heat flux | W m$^{-2}$ | 30 min (average) | -15% to +5% in most common soils |


**Table 3. Instrument details for the SAJESS MUST Trailer site**

| Installation (Instrument abbr.) | Instrument | Installation height | Variable | Units | Resolution | Accuracy |
|---|---|---|---|---|---|---|
| Met tripod (MET) | Campbell Scientific CR1000X datalogger | 1.5 m | data recording | NA | NA | NA |
| | Vaisala HMP155 Temperature/RH probe | 2.00 m | Air temperature Relative humidity | °C % | 1 min (average) | 0.226+0.0028×reading (−80 to +20 °C) 0.055+0.0057×reading (+20 to +60 °C) |
| | Kipp & Zonen CNR4 Net radiometer | 1.80 m | Net 4-way radiation | W m$^{-2}$ | 1 min (average) | < 5 % |
| | Campbell Scientific SR50A | 1.80 m | Snow depth | m | 1 min (average) | ±1 cm |
| 10 m Mast (MAST) | Campbell Scientific CR1000X datalogger | 1.5 m | data recording | NA | NA | NA |
| | Pond engineering K63 Hotplate precipitation gauge | 2.60 m | Air temperature Barometric pressure Precipitation rate Accumulation Windspeed Hotplate power | °C kPa mm hr$^{-1}$ mm m s$^{-1}$ W | 1 min (average) 5 min (average) | ±1 °C ±1 kPa ±0.5 mm hr$^{-1}$ ±0.5 mm ±1 ms$^{-1}$ NA |
| MASC Platform (MASC) | Multi-angle snowflake camera | 1.00 m | Series of 3 images | NA | Up to 3 Hz | NA |
| MRR Pro (MRR) | METEK MRR Pro | 1.30.m | Doppler raw spectra Reflectivity (Ze) Doppler velocity ($W$) Spectral width (σ) | dB dBz m s$^{-1}$ m s$^{-1}$ | see Section 3.1.5 | 0.53 dB 0.53 dBZ 0.109 ms$^{-1}$ 0.09 ms$^{-1}$ |
| Macrophotography (MP) | Nikon D80 with 60 mm macro lens | NA | Series of 9 images | N/A | 10 min | NA |

| | | | Sky condition<br>Cloud type<br>Precipitation type<br>Blowing snow<br>Light precipitation | Oktas<br>Type<br>Type<br>Y/N<br>Y/N | | |
|---|---|---|---|---|---|---|
| Manual observations (OBS) | Volunteer and/or student | NA | Sky condition<br>Cloud type<br>Precipitation type<br>Blowing snow<br>Light precipitation | Oktas<br>Type<br>Type<br>Y/N<br>Y/N | 10 min | NA |
| Upper air observations (SB) | iMet-3050A 403 MHz portable sounding system with iMet-4 radiosonde | NA | Air temperature<br>Relative humidity<br>Wind speed<br>Wind direction<br>Pressure<br>Geopotential height | °C<br>%<br>m s$^{-1}$<br>degrees<br>hPa<br>m | 1 sec | ± 0.5 - 1.0 °C<br>± 5%<br>± 0.5 m s$^{-1}$<br>± 1 degrees<br>± 0.5 - 2.0 hPa<br>± 15 m |




**Table 4. Locations of CoCoRaHS stations founded during SAJESS, with HOBO MX2301A Temperature/RH information (if**
**present)**

| CoCoRaHS ID | Temp/RH sensor ID (if present) | Lat (°N) | Lon (°W) | Elevation (m) | Temp/RH sensor period of record |
|---|---|---|---|---|---|
| CAN-NB-111 | SJ_HOBOTEMP_01 | 47.37 | 68.32 | 166 | 11/12/2020 – 30/04/2021 |
| CAN-NB-112 | Not present | 47.35 | 68.67 | 198 | |
| CAN-NB-113 | SJ_HOBOTEMP_02 | 47.25 | 68.03 | 155 | 11/12/2020 – 30/04/2021 |
| CAN-NB-114 | SJ_HOBOTEMP_03 | 47.37 | 68.31 | 197 | 10/12/2020 – 30/04/2021 |
| CAN-NB-115 | SJ_HOBOTEMP_04 | 47.43 | 68.39 | 142 | 10/12/2020 – 30/04/2021 |
| CAN-NB-117 | SJ_HOBOTEMP_05 | 47.45 | 68.32 | 326 | 11/12/2020 – 30/04/2021 |
| CAN-NB-121 | Not present | 47.21 | 67.96 | 154 | |
| CAN-NB-122 | Not present | 47.40 | 68.34 | 144 | |
| CAN-NB-126 | Not present | 47.38 | 68.31 | 192 | |
| CAN-NB-127 | SJ_HOBOTEMP_06 | 47.36 | 68.16 | 176 | 05/03/2021 – 30/04/2021 |
| CAN-NB-133 | SJ_HOBOTEMP_07 | 47.35 | 68.46 | 332 | 05/03/2021 – 30/04/2021 |
| CAN-NB-134 | Not present | 47.28 | 68.41 | 155 | |
| CAN-NB-135 | SJ_HOBOTEMP_08 | 47.26 | 68.61 | 167 | 12/12/2020 – 30/04/2021 |
| CAN-NB-139 | SJ_HOBOTEMP_09 | 47.37 | 68.34 | 241 | 10/12/2020 – 30/04/2021 |
| CAN-NB-140 | Not present | 47.36 | 67.97 | 295 | |
| CAN-NB-141 | Not present | 47.43 | 68.29 | 348 | |
| CAN-NB-142 | Not present | 47.36 | 68.36 | 189 | |
| CAN-NB-143 | SJ_HOBOTEMP_10 | 47.24 | 68.70 | 170 | 12/12/2020 – 30/04/2021 |
| CAN-NB-144 | SJ_HOBOTEMP_11 | 47.37 | 68.28 | 141 | 10/12/2020 – 30/04/2021 |
| CAN-NB-145 | SJ_HOBOTEMP_12 | 47.33 | 68.09 | 231 | 05/03/2020 – 30/04/2021 |
| CAN-NB-147 | Not present | 47.35 | 68.22 | 164 | |
| N/A | SJ_HOBOTEMP_13 | 47.29 | 68.39 | 156 | 05/03/2021 – 30/04/2021 |
