# Peer review of "Atmospheric and surface observations during the Saint John"

_Earth System Science Data, 2023_

## Referee Comment (RC1)

**Review of "Atmospheric and surface observations during the Saint John River Experiment on Cold Season Storms (SAJESS)" by Hadleigh Thompson et al.**

This paper presents a dataset of winter- and springtime measurements in the upper Saint John river basin, on the border of Maine, Quebec and New Brunswick. The data include a collection of surface observations of precipitation and atmospheric variables, as well as certain surface state variables and radar measurements.

The authors provide a clear description of the various instruments and a mostly good overview of how the data were processed. They also discuss shortcomings and challenges that were encountered. The dataset itself is very well documented.

The campaign is interesting and the data will likely foster research on cold season precipitation and snowpack evolution. The manuscript is clearly written. I do have a few concerns regarding how the data are presented and am missing some important information in the manuscript. As a reader, I am still unsure of how much was collected in terms of usable information (see major comment #1), and I feel the authors should provide more perspective on the dataset along this line. I think my comments will be addressed easily, but I do believe they should be taken into account to increase the impact of this study.

Major comments:

1. I am missing an overview of the dataset in terms of the precipitation events that were observed. Somewhere in the manuscript the reader should be able to understand how many precip events were observed / how many hours of precip / how many hours of each precip type (rain, snow, mixed, melting snow...) / number of rain-on-snow events / total precip amount /… If not all of these items, at least several. Including this type of information will help the user understand what possibilities the dataset opens up.

2. Along the same line, the only "overview" figure of the data (Fig. 7) is quite basic (and could probably have been derived from measurements of a usual weather station). It would be helpful to have a more specific overview figure from the campaign (e.g., from the IOP) that illustrates some of the unique features of the dataset. I can understand if the data have not all been processed to the point that statistics can be derived from them – and this paper is not intended to present a scientific analysis –, but I believe the authors could still give more flavor to the data overview (e.g., with another timeseries or a pie chart from a relevant instrument). For example, in the introduction, the authors mention the relevance of the campaign for the study of how precipitation impacts the snowpack evolution, but this is not put forward in any of the first results / data preview sections. Other example: measurements of precipitation phase seems to be a crucial aspect of the campaign, but are barely put forward (except in half a sub-panel of Fig. 8).

3. In the data description section, nothing is said about the SRS and CoCoRaHS observations. There is only a sentence in the last section to highlight the challenges that were encountered – even though the abstract highlights them as an important element of the setup. It would be nice to have more information on the data that were collected through these platforms, even if they are not perfect. Consider including another figure (or subpanel to an existing figure) showcasing these measurements. Sec. 3.4 introduces nicely the set-up and suggests that bi-dimensional precipitation fields can be reconstructed, and that information on liquid content is available, so I was disappointed not to find any follow-up on these aspects in the later sections.

4. The introduction would benefit from a broader perspective – were similar set-ups deployed somewhere else in the world? l. 58 "no studies of storms and precipitation (…) have been conducted in this region" - how about in other regions? This could also be the place to highlight possible specificities of the SAJESS campaign (in terms of objectives, instrumentation, strategy…)

besides the geographical location, in comparison with previous field deployments (e.g., GCPEX, OLYMPEX...)

5. I am slightly confused with the processing of disdrometer data (Sec 3.1.4, 4.2.4, Fig. 8, Table 2). Did you implement a particle type classification on your disdrometer data? This is not mentioned in the text and does not seem to be in the dataset, but you show it on Fig. 8. In the description of Fig. 8 (either in the text or in the caption), include which classification algorithm was used.  If the info on particle type is not shared in the dataset, I find misleading the sentence of the Data processing section 4.2.4 "Our primary use of Parsivel disdrometer data concentrates on the diagnosis of hydrometeor phase and type". Also, in the dataset I find the variables "Intensity of precipitation" and "Snow intensity" - is the former only liquid? What processing is applied here to derive these variables, is this the built-in software? As the disdrometer is an essential component of the setup, the variables shared in the dataset deserve a more clear description in my opinion.

6. Sec. 3.2 Macrophotographs: please explain the strategy adopted here. In Fig. 9 you mention a felt pad, this would be worth including in the text. Also, mention if the pad is wiped after taking a picture, if the camera is always at the same position, etc.

7. This may be less critical, but the summary section is very short. It would be beneficial to end with some more insights on the scientific questions which the dataset allows addressing.

8. I believe that during campaigns aiming at studying the snowpack evolution, it is frequent to measure the snow water equivalent (SWE). This is not the case here, perhaps for practical reasons or because no such instrument was available. It may be worth discussing this in the manuscript, perhaps in the "Challenges" section?

Minor comments
1. l. 16 Mention in the abstract where the St John River basin is in the world
2. l. 31 Already in the abstract you could include a brief statement to show the outcome of the campaign (e.g., number of precip hours …)
3. Fig. 1 :
   ○ Include a scale in the zoomed box (Edmunston)
   ○ Operational weather radars are shown but never mentioned in the text. This is misleading since you have MRRs in the two main sites.
4. l. 63 To improve clarity, I recommend ending the introduction with a brief outline of the structure of the manuscript. Otherwise it is not very clear that you choose to first describe the sites, then the instrumentation.
5. Sect 2.1 It would be interesting to put some perspective here to explain the choice of the location. At least, a brief description of the winter climatology in these sites would be beneficial.
6. l. 98 "The identification of precipitation phase was achieved by the installation of a K63 Hotplate...". Unless I am mistaken, the K63 hotplate measures the liquid-equivalent precipitation rate and accumulation, but does not discriminate phase. In the "Precipitation phase observatory", only the disdrometer is able to provide information on precipitation phase at the ground, correct? (The MRR somehow indirectly, I suppose). This sentence is somehow misleading.
7. l.125 "At several sites" → state how many sites.
8. Sect. 2.4 and Sect. 3.5: Since the CoCoRAHS measurements are not included in the dataset you provide but should be accessed from the website, consider including in Appendix or in the dataset the list of station numbers / names from SAJESS. As there are numerous stations available on the website, it is cumbersome for the user to look for those of the campaign

without more detailed info. I tried filtering the names with "SaJESS" but this did not seem to work.

9. Sec. 3.1.3 Mention that the hotplate measures equivalent liquid precip
10. Sec. 3.1.6 Mention that the MASC cameras capture images of a particle from different angles. Also, the last sentence of this paragraph is a bit ambiguous as to whether or not these algorithms were implemented on your data, or if the user should implement them.
11. Sec. 3.1.7 Out of curiosity, would it be possible to identify the times when this happened from the altitude sensor on the instrument? Or are the measurements too noisy?
12. l. 219 "Lachapelle and Thériault, 2021a" : reference is missing.
13. Sec. 3.3 It would be interesting to show the dates when radiosoundings are available (e.g., in Fig. 6 or 7), if not too heavy.
14. l. 339 Are these dates flagged somewhere in the dataset or readme files? I couldn't find.
15. Fig. 6: It is hard to see the dates (where is the beginning of April?), the reader has to count days from beginning of March.
16. Sec 4.2.5 You mention comparisons with Caribou. In Fig. 1 you also show another upper air station SE of Fredericton, did you use measurements from this station? Otherwise you may consider removing it from Fig. 1
17. Sec 4.3.1 This could be the place to expand on the conditions that were observed during the campaign (see major comment #1). You could also include the dominant synoptic conditions during the winter and spring, …
18. Sec. 4.3.3 Include a few words on the context of this snowfall event (similar to the beginning of 4.3.2). You may for instance mention temperatures at the site, precip rates, …
19. Fig. 9 – 10. It is a pity that those not very quantitative figures are the only data shown from the IOP (cf. major comment #2), which has a very nice setup.
20. Table 2 and 3: for the instruments that require calibration (e.g., on the met or flux tripods, radars,..) it would be good to include some information on the quality of the calibration (e.g., date of the last calibration, …)
21. Table 2 and 3: What should be in the "Variable" column? Raw variables measured by the sensor, or processed variables available in the files? There seem to be some inconsistencies (e.g., only 2 variables for the Parsivel, but 4 variables for the MRR).
22. Table 2 and 3, MRR
    ○ Resolution: include range resolution + velocity range (min and max vel) + vel. resolution
    ○ Accuracy: include sensitivity information (e.g., min. reflectivity detected at 1km range)
23. Table 3, Met tripod SR50: "snow depth". If I understood correctly, snow is removed after a snowfall event, so perhaps "fresh snow depth" would be more accurate.

Technical corrections
l. 29 Should be "Thompson et al., 2023"
l. 37 "Bay of Fundy (Fig.1)": add the Bay of Fundy to Fig. 1
l. 37 "It covers...": unclear what "it" refers to. Replace with "The SJ river watershed" / "The basin"…
l. 54 Budhathoki et al. (2022): reference is not included
l. 101 "information on the atmospheric conditions aloft": this is not very precise and slightly misleading, the radar informs on precipitation aloft (not all atmospheric conditions)
l. 119-120 "focus on observations, rather than instrumentation": unclear
l. 205 "Garrett"
l. 220 "SLR": don't think the acronym was defined.
l. 280 "in-situ": not sure this is the best word.
l. 321 "Although": I don't understand the use of this word here.
l. 329 "SR50A": this was not defined.
l. 373 "GEONOR": not defined.

l. 377 is the bias with the T5 or the T1 data? Unclear from the phrasing.

Fig. 8b: A narrower color scale would be relevant as all the data seems to be within 5 dBZ and 30 dBZ. Perhaps a continuous colorbar would also be helpful (as in panel c).

Comments on the dataset

I would like to acknowledge the efforts put into the documentation of the data. The Readme files are extremely clear and this is of great help to a new user of the dataset. Below is a list of minor issues:

- there are some discrepancies in the naming of the files inside the readme files (for example in the SJ_FS_FLUX_readme, l. 134 – 136)
- OTT Parsivel: I could not find values of the diameter and velocity bins
- MRR-PRO: I may have missed but could not find values of the velocity bins of the spectra (if they are indeed missing, one possibility would be to include the bounds + velocity resolution in the readme file so the user can reconstruct this array).
- Macrophotography: Can the user find a scale for the pictures?

---

## Community Comment (CC1)

Review of the paper "Atmospheric and surface observations during the Saint John River Experiment on Cold Season Storms (SAJESS)" by Thompsonet al. submitted to Earth System Science Data (ESSD).

This paper provides a dataset for studying rainfall-on-snow. I think the dataset is valuable, especially for modelers, and the manuscript has been well written. I just have a few comments:

1. Since soil moisture and temperature were measured at the Flux tripod, could you please provide soil types of the site at different layers. It could be useful in modeling.
2. Line 114: you may want to remove "(a)" in the line.
3. From Figure 3, it seems the instruments at MUST are very close to a group of trees. Could you please give more information on the impact of trees on the observations.
4. Likely, this dataset is going to be used by modelers, which is a group of people who do not have rich knowledge on field instruments. Thus, could you please specifically point out what types of data are collected by different instruments? For example, I could not find the descriptions on what variables were being measured by the hotplate precipitation gauge, micro rain radar, multi-angle snowflake camera?
5. Line 211: You mentioned re-extending the 10-m mast several times. Could you please include such information in your data, say, make markers on the date of re-extending.
6. Line 299-305: if you have any reasons for the consistent bias?

---

## Author Response (AR1)

**Reviewer 1:**

**CC1: ['Comment on essd-2023-59'](), Siwei He, 26 Apr 2023:**

Review of the paper "Atmospheric and surface observations during the Saint John River Experiment on Cold Season Storms (SAJESS)" by Thompsonet al. submitted to Earth System Science Data (ESSD). This paper provides a dataset for studying rainfall-on-snow. I think the dataset is valuable, especially for modelers, and the manuscript has been well written.

**I just have a few comments:**

1. Since soil moisture and temperature were measured at the Flux tripod, could you please provide soil types of the site at different layers. It could be useful in modeling.

**Response:** A soil pit analysis was not performed during the installation, and the bulk density variable was left as the default value (1300 kg m$^{-3}$) in the program.

A sentence has been added to Section 4.4 that now reads: *"Due to the lack of soil analysis, the datalogger program's default bulk density value of 1300 kg m$^{-3}$ was used. Although future deployments should not underestimate the time and energy required to install the flux instrumentation correctly, we have retained the flux data as thorough post-processing has allowed for effective analysis of the data (Leroux et al, 2023)."*

2. Line 114: you may want to remove "(a)" in the line.

**Response:** Thank you, this has been removed as suggested.

3. From Figure 3, it seems the instruments at MUST are very close to a group of trees. Could you please give more information on the impact of trees on the observations.

**Response:** We agree that the proximity to trees is less than ideal, and in particular this may have had a significant effect on the wind data recorded at the MUST Trailer (along with the mast extension problems). Therefore, we removed the wind data from the repository dataset.

With respect to other instrumentation, the trees are to the north of the row of instruments, eliminating shading of the tripods and we believe the the effect on air temperature, RH, radiation, the MRR, and the MASC to be negligible. There is a possibility that wind driven snow from the north would be inhibited by the trees, and therefore could result in lower snow amounts on the snow board, although this is not evident on the timelapse images.

As noted in Section 2.3, the primary use of the MUST Trailer location was for manual observations, hydrometeor photography, and balloon releases. The Fixed Station provided a more suitable site for undisturbed near-surface instrumentation.

4. Likely, this dataset is going to be used by modelers, which is a group of people who do not have

rich knowledge on field instruments. Thus, could you please specifically point out what types of data are collected by different instruments? For example, I could not find the descriptions on what variables were being measured by the hotplate precipitation gauge, micro rain radar, multi-angle snowflake camera?

**Response**: All variables recorded by the instruments are listed in Tables 2 and 3. These variables are further documented in the readme files available in the repository, which also include information about the instruments at each site.

5. Line 211: You mentioned re-extending the 10-m mast several times. Could you please include such information in your data, say, make markers on the date of re-extending.

**Response:** Please see our response above regarding wind data. Additionally, wind data from the Environment and Climate Change Canada site located adjacent to the Fixed Station are available online.

6. Line 299-305: if you have any reasons for the consistent bias?

**Response:** Thank you for the question. The sensor was wired to the datalogger improperly, which in-turn caused the datalogger program to interpret the voltage signal incorrectly. This is mentioned in Section 4.2.1.

**Please note:** Any changes to the dataset will be completed and made public at the end of the peer review process.

Reviewer 2:

**RC1: 'Comment on essd-2023-59', Anonymous Referee #1, 30 May 2023**

Review of "Atmospheric and surface observations during the Saint John River Experiment on Cold Season Storms (SAJESS)" by Hadleigh Thompson et al. This paper presents a dataset of winter- and springtime measurements in the upper Saint John river basin, on the border of Maine, Quebec and New Brunswick. The data include a collection of surface observations of precipitation and atmospheric variables, as well as certain surface state variables and radar measurements. The authors provide a clear description of the various instruments and a mostly good overview of how the data were processed. They also discuss shortcomings and challenges that were encountered. The dataset itself is very well documented. The campaign is interesting and the data will likely foster research on cold season precipitation and snowpack evolution. The manuscript is clearly written. I do have a few concerns regarding how the data are presented and am missing some important information in the manuscript. As a reader, I am still unsure of how much was collected in terms of usable information (see major comment #1), and I feel the authors should provide more perspective on the dataset along this line. I think my comments will be addressed easily, but I do believe they should be taken into account to increase the impact of this study.

**Major comments:**

1. I am missing an overview of the dataset in terms of the precipitation events that were observed. Somewhere in the manuscript the reader should be able to understand how many precip events were observed / how many hours of precip / how many hours of each precip type (rain, snow, mixed, melting snow...) / number of rain-on-snow events / total precip amount /… If not all of these items, at least several. Including this type of information will help the user understand what possibilities the dataset opens up.

**Response:** Thank you for the suggestion. We have renamed Section 4.3.1 from *Winter 2020/21* to *Overview*. We now include information about total precipitation observed, maximum snow depth measurement at CoCoRaHS stations, the number of hours of manual observations, and the precipitation types observed. To supplement the text, we have enhanced Figure 7, separating the data into three subplots (air temperature, precipitation, and snow depth/snow on the ground). The figure now also includes measurements taken by the 21 CoCoRaHS observers listed in Table 4. We agree that these changes provide a more complete picture of the conditions observed during the campaign.

The two introductory paragraphs for this section now read:
*"Data covering 1 December 2020 – 30 April 2021 are provided by the instruments at the Fixed Station, which can be supplemented by the near-by ECCC station. Maximum snow depth measured at the Fixed Station was 65 cm, with snowfall amounts comparing well in timing, yet slightly less in magnitude (around 10%), than measured by the ECCC instruments (Figure 7). Maximum snow depth observed at 21 surrounding CoCoRaHS stations (Table 4) ranged from 50 cm to 79 cm, with an*

*average of 54 cm. During this period, the ECCC Geonor measured a total of 287 mm of (liquid equivalent) precipitation, less than the climatology (1993-2022) mean of 364 mm.*

*During the IOP (8 March 2021 - 30 April 2021) manual observations were made during 13 storms (Fig. 6), for a total of 183 hours of precipitation-type observations. Observers were present for 93 hours of snow (8 storms), 63 hours of rain (5 storms), and 27 hours of mixed precipitation (more than one phase of precipitation occurring at the same time; 3 storms). Two storms (12 March 2021; 4-5 April 2021) included rain-snow transitions. Observers took a total of 4483 images of precipitation particles during the IOP. Using the ECCC Geonor, observers were present for 97 mm of the 105 mm of (liquid equivalent) precipitation that fell during the IOP period."*

2. Along the same line, the only "overview" figure of the data (Fig. 7) is quite basic (and could probably have been derived from measurements of a usual weather station). It would be helpful to have a more specific overview figure from the campaign (e.g., from the IOP) that illustrates some of the unique features of the dataset. I can understand if the data have not all been processed to the point that statistics can be derived from them – and this paper is not intended to present a scientific analysis –, but I believe the authors could still give more flavor to the data overview (e.g., with another time series or a pie chart from a relevant instrument). For example, in the introduction, the authors mention the relevance of the campaign for the study of how precipitation impacts the snowpack evolution, but this is not put forward in any of the first results / data preview sections. Other examples: measurements of precipitation phase seems to be a crucial aspect of the campaign, but are barely put forward (except in half a sub-panel of Fig. 8).

**Response:** Thank you for the suggestion. In addition to the changes mentioned above in response to point #1, we have also added text to Section 4.3.2 detailing the effect of the rain-on-snow event on the evolution of the seasonal snowpack. The time and date of the example remains the same, but more information about the measurements recorded during the event have been added.

The section now reads: *"Focused examples of the meteorological measurements at the fixed station are given Fig. 8. This case covers the period 25 – 27 December 2020 during which 22.5 hours of rain-on-snow (14 mm) from 1230 UTC 25 December to 1100 UTC 26 December 2020, with temperatures of 5°C – 6°C, reduced the Fixed Station snow depth to 0.0 m. This was followed closely by a decrease in temperature to < 0°C and ~4.5 hours of snow (4 cm) from 1630 UTC to 2050 UTC 26 December 2021), restarting the seasonal snowpack from bare conditions. Precipitation type derived from the disdrometer data correlates well with the decrease in air temperature below 0°C. The time series29 of particle type can be constructed from the binned fall speed and diameter measurements and by using equations in the literature such as Rasmussen et al. (1999) (also see Ishizaka et al., 2013), or by using the precipitation type codes produced by the instrument software (Table 2). MRR data show the reduction of melting height (determined by the sharp vertical gradient of reflectivity) from ~ 3 km AGL to just over ~ 2 km AGL during this same period. These insights to the evolution of the seasonal snowpack are rare as many standard weather or climate stations may not have the ability to explicitly identify precipitation type."*

For brevity we have decided against including another figure to illustrate these points, relying instead on the revision of Figure 7 and the text to communicate the information.

3. In the data description section, nothing is said about the SRS and CoCoRaHS observations. There is only a sentence in the last section to highlight the challenges that were encountered – even though the abstract highlights them as an important element of the setup. It would be nice to have more information on the data that were collected through these platforms, even if they are not perfect. Consider including another figure (or subpanel to an existing figure) showcasing these measurements. Sec. 3.4 introduces nicely the set-up and suggests that bi-dimensional precipitation fields can be reconstructed, and that information on liquid content is available, so I was disappointed not to find any follow-up on these aspects in the later sections.

**Response:** For CoCoRaHS data we have added Section 4.2.6 (CoCoRaHS precipitation reports). This section outlines the number of measurements taken by CoCoRaHS observers during SAJESS, and refers to the updated Figure 7 that displays CoCoRaHS observations alongside the FIxed Station and ECCC instruments.

The paragraph now reads: *"CoCoRaHS volunteers conducted most of their observations during the IOP, due to the delays in obtaining equipment and training due to public health measures in place at the time. Despite this, 21 CoCoRaHS stations (Table 4) reported 1715 daily total precipitation (liquid equivalent) measurements, 1729 new snow measurements, and 1123 total snow measurements (Fig. 7). A comparison of the daily precipitation and new snow amounts has not been conducted, but the total snow (i.e., snow on the ground) compares well with the Fixed Station and ECCC instruments. CoCoRaHS data are not included in the SAJESS dataset as: (a) CoCoRaHS already provide long-term storage and data retrieval via its own website, and (b) the authors encourage users to explore the CoCoRaHS database for stations that may be useful for their own analysis that were not associated with SAJESS."*

Section 3.4 has been revised to clarify the experimental nature of the SRS in mixed-phase precipitation. Work with these data is ongoing and will hopefully be part of a future SAJESS citizen-science oriented manuscript.

4. The introduction would benefit from a broader perspective – were similar set-ups deployed somewhere else in the world? l. 58 "no studies of storms and precipitation (…) have been conducted in this region" - how about in other regions? This could also be the place to highlight possible specificities of the SAJESS campaign (in terms of objectives, instrumentation, strategy…) besides the geographical location, in comparison with previous field deployments (e.g., GCPEX, OLYMPEX...)

**Response:** Thank you for the suggestion. We have added the following paragraph to the introduction to provide broader context for SAJESS and the techniques employed:

*"There have been several previous cold-season precipitation studies in North America. These include a comparison between orographic winter storms in the San Juan Mountains and Sierra Nevada by Marwitz (1986), the Canadian Atlantic Storms Program (Stewart et al., 1987; Stewart, 1991), that investigated the synoptic and mesoscale structure of Canadian East Coast winter storms, and field-campaigns such as the meteorological monitoring network established for the Vancouver 2010 winter Olympics (Joe et al., 2014), and the Olympic Mountains experiment (OLYMPEX) that studied the modification of Pacific storms by coastal mountain ranges (Houze et al., 2017). Notably, the*

*Vancouver Olympics network utilized a first-generation hotplate precipitation gauge, and while OLYMPEX used similar instrumentation to SAJESS such as disdrometers, weighing rain gauges, and micro rain radars, they also employed instrumented aircraft and a greater range of radar options. Ongoing research into East Coast snowstorm-producing cyclones is being undertaken by the Investigation of Microphysics and Precipitation for Atlantic Coast-Threatening Snowstorms (IMPACTS) campaign. Similar to OLYMPEX, IMPACTS combines surface observations and measurements with airborne remote sensing instruments. In terms of field projects, SAJESS compares closest to the recent (2019) Storms and Precipitation Across the continental Divide experiment (SPADE) (Thériault et al., 2021a, 2022), adopting the methods of manual observations, macrophotography, and micro rain radar and disdrometer deployment, although we built on SPADE by adding a distributed network of temperature and precipitation measurements, and upper air observations. "*

5. I am slightly confused with the processing of disdrometer data (Sec 3.1.4, 4.2.4, Fig. 8, Table 2). Did you implement a particle type classification on your disdrometer data? This is not mentioned in the text and does not seem to be in the dataset, but you show it on Fig. 8. In the description of Fig. 8 (either in the text or in the caption), include which classification algorithm was used. If the info on particle type is not shared in the dataset, I find misleading the sentence of the Data processing section 4.2.4 "Our primary use of Parsivel disdrometer data concentrates on the diagnosis of hydrometeor phase and type". Also, in the dataset I find the variables "Intensity of precipitation" and "Snow intensity" - is the former only liquid? What processing is applied here to derive these variables, is this the built-in software? As the disdrometer is an essential component of the setup, the variables shared in the dataset deserve a more clear description in my opinion.

**Response:** Thank you for noticing the inconsistencies with our reporting of the disdrometer data. We have amended Table 2 to include all variables contained in the disdrometer files, and updated the text in Sections 3.1.4 and 4.2.4 to remove any ambiguity in what is, and is not, included in the dataset. We have also included the mid-values and spreads for each of the 32 fall speed and 32 size bins the software utilizes to count particles in the readme file. We have updated the caption for Figure 8 to clarify that precipitation types displayed can be determined from the Parsivel data but are not included. Finally, we include a reference for the operating instructions, which we encourage users to refer to.

Section 3.1.4 now reads: *"A laser-optical disdrometer was deployed at the Fixed Station for the duration of the field campaign on the same tripod as the standard meteorological instruments, at 2.8 m AGL (Fig. 3). The disdrometer provides measurements of the size and speed of falling hydrometeors which, when post-processed accordingly, can provide a classification of hydrometeor type (Hauser et al., 1984; Rasmussen et al., 1999; Löffler-Mang and Joss, 2000; Ishizaka et al., 2013; Thériault et al., 2021a). Included in the dataset are variables derived by the manufacturer's software such as precipitation intensity (mm h$^{-1}$), number of detected particles, and several national standard present weather codes. The spectrum of particles is provided by 1024 columns representing the 32 × 32 matrix of particle fall speed and diameter bins. The mid-values and spread of these bins are listed in the readme file supplied with the dataset and in the disdrometer user manual (OTT, 2019)."*

The first sentence in Section 4.2.4 now reads: "The disdrometer dataset contains 1024 columns, which includes  total particle counts and fallspeed, particle spectrum (Section 3.1.4) and other derived

variables listed in Table 4. This configuration option is provided by the OTT software as raw data. At present we can only comment on a preliminary analysis of the OTT-derived precipitation intensity."

6. Sec. 3.2 Macrophotographs: please explain the strategy adopted here. In Fig. 9 you mention a felt pad, this would be worth including in the text. Also, mention if the pad is wiped after taking a picture, if the camera is always at the same position, etc.

**Response:** Thank you for your suggestion, we have added a sentence to Section 3.2 that reads: *"Precipitation was collected on a black velvet collection pad during a short period (5 - 30 seconds), depending on the precipitation rate. The aim was to ensure that there were enough particles to represent the precipitation conditions, while avoiding particles overlapping each other. The collection pad was wiped every time after the series of nine images were taken using a set sequence of movements of the camera upon a sliding frame. Images of a ruler placed across the pad for scale were taken intermittently, for example when the camera was removed/replaced on the stand. Users can assume the scale remains constant between these images (each image is 2 cm across). The scale images are listed in the macrophotography readme file. More details of the method, including images of the equipment, can be found in Thériault et al. (2021a) and Thériault et al. (2018)."*

7. This may be less critical, but the summary section is very short. It would be beneficial to end with some more insights on the scientific questions which the dataset allows addressing.

**Response:** Thank you for your suggestion. With the comments from all four reviewers we have revised the summary, and it now reads:
*"A valuable dataset was collected during the 2020-2021 Saint John River Experiment on Cold Season Storms (SAJESS) over Eastern Canada. The dataset contains automatic measurements, manual observations, and photos of the meteorological conditions and precipitation at the surface, micro rain radar measurements, upper air observations, and measurements of precipitation amounts by community volunteers.*

*The experiment included an intensive observation period during March – April 2021, to document conditions during the melt season with its below-average snowpack melt a month earlier than normal. The study region and downstream communities are historically prone to ice jams and flooding during late winter and early spring, which can be influenced by air temperature, rain-on-snow events, and the overall evolution of the seasonal snowpack. Despite these hazards being minimal during the campaign, the dataset will contribute to an understanding of snowpack evolution across the region by a variety of mechanisms. Firstly, the spatially distributed measurements from CoCoRaHS observers and SRS installations give a broader view of precipitation types and amounts than the typical isolated weather or climate station. The dataset contains more than 1700 measurements from these community volunteers, and the experiment doubled as a learning opportunity for the classes of school children that contributed by recording snow board and rain gauge measurements. Secondly, the identification of precipitation type at the surface, particularly when there is snow on the ground and temperatures are near 0°C, can be vital for forecasting the resulting snowpack and hydrological response. This was achieved during SAJESS by combining high temporal resolution measurements (1-minute) of precipitation amount, snow depth, and disdrometer data, with manual observations and photographic imagery of hydrometeors at the surface. Verifying the precipitation type using these measurements removes the uncertainty of precipitation type estimation that is necessary without such*

*an array of measurements. Thirdly, observations of atmospheric conditions and precipitation aloft from the micro rain radar and upper air soundings will advance our knowledge of pre-storm conditions and the influence of tropospheric conditions on the precipitation type observed at the surface. When these conditions result in solid precipitation at the surface, inspection of the macrophotographs and MASC images can be included to confirm particle type, size, and shape, and for snow, crystal habit and riming condition. The SAJESS dataset provides an opportunity to investigate the micro-scale influences these differing particle types and crystal habits have on snowpack evolution.*

*In addition, the SAJESS dataset provides the opportunity to advance forecasting and model evaluation. The breadth of instruments and human observations quantifies many parameters not archived in traditional meteorological databases. Examples of this include the non-standard timing of balloon launches, spatially distributed precipitation and temperature observations, as well as energy balance and flux data. The identification of particle type obtained by using non-standard observations such as the MASC provides ground truth to help further model microphysics.*

*Finally, the SAJESS field campaign also highlights the need to enhance measurements of precipitation and snow in the upper Saint John River Basin. Near real-time access to data during the campaign allowed for the monitoring of meteorological conditions as they occurred, which can be vital for emergency management, ice jam and flood forecasting, and river navigation. This is not usually available in areas where weather or climate stations are sparse. We envision that consideration be given to furthering the meteorological monitoring network in the upper Saint John River Basin to increase the resolution and frequency of measurements and to better anticipate ice jams and major flooding events along the Saint John River."*

8. I believe that during campaigns aiming at studying the snowpack evolution, it is frequent to measure the snow water equivalent (SWE). This is not the case here, perhaps for practical reasons or because no such instrument was available. It may be worth discussing this in the manuscript, perhaps in the "Challenges" section?

**Response:** SWE measurements are an integral part of data collected by CoCoRaHS observers, as mentioned in Section 3.5.1. This information is available via the CoCoRaHS website and we have updated Figure 1 and Table 4 to indicate the locations of the CoCoRaHS stations implemented during the SAJESS campaign. The CoCoRaHS data reduced the necessity for SAJESS team members to make SWE measurements, especially during periods of restrictive public health measures.

**Minor comments**

1. l. 16 Mention in the abstract where the St John River basin is in the world

**Response:** Thank you for the suggestion, the abstract now contains the sentence: *"To study the impact of winter and spring storms on the snowpack in the upper Saint John River (SJR) basin. The region encompasses parts of the US state of Maine (ME) and the Canadian provinces of Quebec (QC) and New Brunswick (NB), the Saint John River Experiment on Cold Season Storms (SAJESS) utilized …"*

2. l. 31 Already in the abstract you could include a brief statement to show the outcome of the

campaign (e.g., number of precip hours …)

**Response:** We have revised the abstract to now include a brief statement as suggested. The section now reads: *"A lower-than-average snowpack peaked at ~65 cm, with a total of 287 mm of precipitation (liquid equivalent) falling between December 2020 and April 2021, a 21% lower amount of precipitation than the climatological normal. Observers were present for 13 storms, conducting manual observations for 183 hours of precipitation, while taking more than 4000 images of hydrometeors. The inclusion of local volunteers and schools provided an additional 1700 measurements of precipitation amounts. The resulting datasets include optical disdrometer data, micro rain radar output, near-surface meteorological observations, as well as temperature, pressure, humidity, and precipitation data."*

3. Fig. 1 : ◦ Include a scale in the zoomed box (Edmunston) ◦ Operational weather radars are shown but never mentioned in the text. This is misleading since you have MRRs in the two main sites.

**Response:** Thank you for your suggestion. A scale bar has been added to the Edmundston inset and the weather radars removed from the map.

4. l. 63 To improve clarity, I recommend ending the introduction with a brief outline of the structure of the manuscript. Otherwise it is not very clear that you choose to first describe the sites, then the instrumentation.

**Response:** Thank you for the suggestion. The last paragraph of the introduction now reads:

*"The SAJESS dataset contains meteorological and precipitation data that were collected at a fixed station from 1 December 2020 until 30 April 2021, and an intensive observation period (IOP) that took place from 8 March to 30 April 2021. The objective of this paper is to describe the data collected during SAJESS, provide examples of the measurements and how they can be combined to broaden the picture of meteorological conditions observed, and to illustrate to stakeholders and potential users the importance of the field campaign and dataset.*

*The paper is organized as follows: the sites used during SAJESS are described in Section 2; the instruments, manual observations, and other sources of data are detailed in Section 3; an overview of data processing, management, and validity, along with examples of data collected throughout SAJESS are presented in Section 4; and a summary, including a discussion on potential future analyses, is given in Section 5."*

5. Sect 2.1 It would be interesting to put some perspective here to explain the choice of the location. At least, a brief description of the winter climatology in these sites would be beneficial.

**Response:** We have detailed a few justifications for the choice of location in the introduction (e.g., area is prone to flooding, paucity of data in the area). For brevity, we feel the manuscript should focus on describing the instrumentation and data obtained during the campaign. Figure 7 has been revised to provide a greater overview of the climatology of the study sites, and measurements at the Fixed Station are now compared with climatological normals in the text.

6. 98 "The identification of precipitation phase was achieved by the installation of a K63 Hotplate...". Unless I am mistaken, the K63 hotplate measures the liquid-equivalent precipitation rate and accumulation, but does not discriminate phase. In the "Precipitation phase observatory", only the disdrometer is able to provide information on precipitation phase at the ground, correct? (The MRR somehow indirectly, I suppose). This sentence is somehow misleading.

**Response:** The sentence has been reworded to remove any ambiguity, and now reads: "The identification of precipitation phase was achieved at the Fixed Station by the installation of a laser-optical disdrometer for recording particle size and fall speed, and by a vertically pointing micro rain radar (MRR) to provide information on the atmospheric conditions aloft (see Section 3 and Table 2). A K63 Hotplate Total Precipitation Gauge (henceforth, hotplate) was installed to measure precipitation rate and amount."

7. 125 "At several sites" → state how many sites.

**Response:** Six sites - this has been updated in the text as suggested.

8. Sect. 2.4 and Sect. 3.5: Since the CoCoRAHS measurements are not included in the dataset you provide but should be accessed from the website, consider including in Appendix or in the dataset the list of station numbers / names from SAJESS. As there are numerous stations available on the website, it is cumbersome for the user to look for those of the campaign without more detailed info. I tried filtering the names with "SaJESS" but this did not seem to work.

**Response**: Table 4 lists the stations established with assistance from SAJESS staff, complete with the location, elevation, and CoCoRaHS stations number. These are also now shown in Figure 1.

9. Sec. 3.1.3 Mention that the hotplate measures equivalent liquid precip

**Response:** Thank you for the suggestion, we have added a sentence to Section 3.1.3 that reads: *"The hotplate measures liquid-equivalent precipitation by recording 1-minute and 5-minute running averages of precipitation rate and wind speed (needed as control for precipitation rate). An accompanying environmental sensor measures air temperature, RH, and atmospheric pressure with the same 1-minute and 5-minute temporal resolution."*

10. Sec. 3.1.6 Mention that the MASC cameras capture images of a particle from different angles. Also, the last sentence of this paragraph is a bit ambiguous as to whether or not these algorithms were implemented on your data, or if the user should implement them.

**Response:** Thank you for the suggestion, the sentence in Section 3.1.6 now reads: *"The MASC consists of three high-speed cameras housed in a single enclosure, with 36° separation between each camera, and a focal point ~10 cm for each lens. The cameras take images simultaneously when particles are detected within a ring-shaped viewing area (Fig. 4c inset). The three images from each trigger consequently show the hydrometeor(s) from slightly different angles."*

The final sentence of Section 3.1.6 has been removed to avoid ambiguity, as suggested.

11. Sec. 3.1.7 Out of curiosity, would it be possible to identify the times when this happened from the

altitude sensor on the instrument? Or are the measurements too noisy?

**Response:** We have excluded the wind data from the dataset due to: (a) issues with the mast staying extended, and (b) the proximity of the anemometer to the nearby building and trees. Wind data from the ECCC wind monitor atop the 10-m mast next to the Fixed Station are available from the ECCC.

12. l. 219 "Lachapelle and Thériault, 2021a" : reference is missing.

**Response:** Thank you for noticing this inconsistency. The citation has been corrected to read *'Thériault et al., 2021a'.*

13. Sec. 3.3 It would be interesting to show the dates when radiosoundings are available (e.g., in Fig. 6 or 7), if not too heavy.

**Response:** Thank you for the suggestion. Although it does not illustrate every attempted sounding, we have added the days with soundings to Figure 6.

14. l. 339 Are these dates flagged somewhere in the dataset or readme files? I couldn't find.

**Response:** These dates are not flagged as they were deduced with a rudimentary method for detecting snow on the sensor window, and we suggest users employ one of the more rigorous approaches suggested in the text (line 340).

15. Fig. 6: It is hard to see the dates (where is the beginning of April?) The reader has to count days from the beginning of March.

**Response:** Thank you for the observation. The x-axis tick marks and labels for Figure 6 have been improved, and they now denote the start and end of each month for the entire campaign, and the start, 15th, and end of each month of the IOP.

16. Sec 4.2.5 You mention comparisons with Caribou. In Fig. 1 you also show another upper air station SE of Fredericton, did you use measurements from this station? Otherwise you may consider removing it from Fig. 1

**Response:** We have removed the upper air stations and the weather radar locations from Figure 1, as they do not make up part of this dataset. We have changed the labelling of 'weather stations' on the map to '*Federal / Provincial Weather Station'* for clarity.

17. Sec 4.3.1 This could be the place to expand on the conditions that were observed during the campaign (see major comment #1). You could also include the dominant synoptic conditions during the winter and spring, …

**Response:** As suggested, Section 4.3.1 has been enhanced with the inclusion of information about the observed conditions (see response to point #1). We have decided against including information regarding the synoptic conditions during the campaign, as they are outside the scope of this manuscript.

18. Sec. 4.3.3 Include a few words on the context of this snowfall event (similar to the beginning of

4.3.2). You may for instance mention temperatures at the site, precip rates, …

**Response:** Thank you for the suggestion. We have added a short quantitative analysis of the conditions observed, and Section 4.3.3 now reads:

*"To complement the automated measurements made at the Fixed Station, the MUST Trailer site provided observations of precipitation type, photographic imagery of particles, and upper air soundings, in addition to the meteorological tripod and hotplate. Images and observations taken during a snow event that occurred on 18 March 2021 are shown in Figs. 9 and 10.*

*From 1000 UTC to 1430 UTC 18 March 2021, light intermittent snowfall with precipitation rates between 0.4 – 0.6 mm h$^{-1}$ resulted in ~1.5 cm of snowfall at the MUST Trailer. Precipitation rates were at times so low that the hotplate did not record any precipitation, but snowfall was confirmed by the manual observations. The Fixed Station disdrometer also detected very light precipitation (maximum rate of 0.6 mm h$^{-1}$) with only 0.2 mm total (liquid equivalent) for the same period. The ECCC GEONOR did not record any precipitation during this time. Air temperature at the MUST Trailer remained between -1°C and 0°C during most of the snowfall, with a short (~30 min) period of snow grains being reported when temperatures increased to 0°C – 0.5°C. Macrophotography images taken from the SAJESS MUST Trailer provide confirmation of hydrometeor type. From these images (Fig. 9) crystal habit, size distribution, and riming can be diagnosed, and they align with the 10-minute observations that recorded overcast skies, nimbostratus clouds, and snow. Also, during this time, the MASC captured images of a variety of hydrometeors, with the clearest images being of large aggregates (Fig. 10a). At 1200 UTC 18 March 2021 a deep saturated layer between 900 and 700 hPa is evident on the upper air observations from the MUST Trailer, matching the vertical profile observed in the National Weather Service (NWS) sounding from Caribou (ME) (Fig. 10b).*

*After a break in conditions, light precipitation made up of wet snow and rain occurred between 1730 UTC – 1950 UTC 18 March 2021. The mixed precipitation erased the snow collected on the snowboard under the MUST Trailer SR50A. This is important as it illustrates what CoCoRaHS observers would have encountered the next morning (19 March 2021), when no observations of new snow were recorded."*

19. Fig. 9 – 10. It is a pity that those not very quantitative figures are the only data shown from the IOP (cf. major comment #2), which has a very nice setup.

**Response:** While we agree a detailed illustration of data captured during the 18 March 2021 would be interesting, Section 4.3.3 is intended to illustrate data obtained at the MUST trailer that is unique to the instrumentation at that site. We have enhanced the text in Section 4.3.3 as point #18 above.

20. Table 2 and 3: for the instruments that require calibration (e.g., on the met or flux tripods, radars,..) it would be good to include some information on the quality of the calibration (e.g., date of the last calibration, …)

**Response:** We have chosen to keep specific information about each instrument, such as last date of calibration, included in the readme documentation with the dataset. A sentence has been added to the end of introduction of Section 3.1 that reads: "Further details of each sensor, such as the date of

manufacture, last calibration, and serial number, are also provided in the readme files."

21. Table 2 and 3: What should be in the "Variable" column? Raw variables measured by the sensor, or processed variables available in the files? There seem to be some inconsistencies (e.g., only 2 variables for the Parsivel, but 4 variables for the MRR).

**Response:** Thank you for alerting us to this inconsistency. These variables are what are available in the data files in the repository (i.e., for many instruments it is unprocessed, except for the MRR which has been processed as mentioned in the text). The disdrometer section of Table 2 has been updated to include all variables in the disdrometer files.

22. Table 2 and 3, MRR ◦ Resolution: include range resolution + velocity range (min and max vel) + vel. resolution ◦ Accuracy: include sensitivity information (e.g., min. reflectivity detected at 1km range)

**Response:** As the space for such details in Tables 2 and 3 is limited, we have added this information to Section 3.1.5 and to the MRR readme files.

Section 3.1.5 now reads: *"A METEK MRR-2 was installed at the Fixed Station, 2.6 m AGL, at the western end of the instrument array (Fig. 3). The MRR-2 has 32 range gates (height steps) and we set the height resolution to the maximum of 200 m, giving a maximum height of 6400 m. Raw data from the MRR-2 were post-processed using the IMProToo algorithm from Mahn and Kollias (2012), as outlined in Thériault et al. (2021a). An MRR-Pro was installed at 1.3 m AGL at the MUST Trailer (Fig. 4d). Settings for the MRR-Pro were: 128 range gates, 64 spectral lines, 10 s sampling, 50 m height resolution, and 6350 m ceiling. The combination of these parameters results in a velocity range of 12 m/s, and a velocity resolution of 0.19 m s$^{-1}$. The 10 s averaging time results in a total of 305 spectra being averaged for each measurement."*

Sensitivity information was not readily available from the material supplied from METEK. As with all instrumentation, we encourage users to refer to the manufacturer's documentation and/or contact them for further information.

23. Table 3, Met tripod SR50: "snow depth". If I understood correctly, snow is removed after a snowfall event, so perhaps "fresh snow depth" would be more accurate.

**Response:** Thank you for the suggestion, but we have decided to keep the naming of variables at the two sites as close as possible. This includes the use of the term *snow depth* for the measurements recorded by the SR50A at each site.

**Technical corrections**

l. 29 Should be "Thompson et al., 2023"

**Response:** This has been corrected as suggested.

l. 37 "Bay of Fundy (Fig.1)": add the Bay of Fundy to Fig. 1

A label for the Bay of Fundy has been added to Figure 1 as suggested.

l. 37 "It covers...": unclear what "it" refers to. Replace with "The SJ river watershed" / "The basin"…

**Response:** This has been corrected by changing the sentence to: *"The Saint John River watershed covers 55,000 km², …"*

l. 54 Budhathoki et al. (2022): reference is not included

**Response:** This reference has been added to the References section.

l. 101 "information on the atmospheric conditions aloft": this is not very precise and slightly misleading, the radar informs on precipitation aloft (not all atmospheric conditions)

**Response:** "atmospheric conditions aloft" has been changed to "precipitation aloft", as suggested.

l. 119-120 "focus on observations, rather than instrumentation": unclear

**Response:** The sentence has been changed to read: *"The proximity of the instruments to the open ponds, nearby railway, and urban environment, resulted in the focus on manual observations of weather conditions and precipitation type, rather than automated instrumentation,...".*

l. 205 "Garrett"

**Response:** This has been corrected.

l. 220 "SLR": don't think the acronym was defined.

**Response:** SLR is now defined as: *"... single lens reflex (SLR)..."*, as suggested.

l. 280 "in-situ": not sure this is the best word.

**Response:** Thank you for the suggestion, the sentence now reads: *"Hourly data from the MUST Trailer radar (MRR Pro) have been archived as .nc files as these data are produced by the instruments embedded processor (METEK, 2017)."*

l. 321 "Although": I don't understand the use of this word here.

**Response:** Thank you for the suggestion, the sentence now reads: *"We found similar results to those of Domine et al. (2021) whereby RMSE between the net-radiometer (using LW↑), and the IRRs…"*

l. 329 "SR50A": this was not defined.

**Response:** This has been defined, the sentence now reads: *"Snow depth measurements from the sonic ranger (SR50A) on the Fixed Station meteorological tripod…"*

l. 373 "GEONOR": not defined.

**Response:** Geonor is the name of the company that manufactures the weighing precipitation gauge, however it is also colloquially used as the name of the gauge. We now define the term Geonor (now lowercase) in Section 2.2 in a sentence that reads: *"Geonor T-200B weighing precipitation gauge with a single alter shield (henceforth Geonor),..."*

l. 377 is the bias with the T5 or the T1 data? Unclear from the phrasing. Fig. 8b: A narrower color scale would be relevant as all the data seems to be within 5 dBZ and 30 dBZ. Perhaps a continuous colorbar would also be helpful (as in panel c).

**Response:** This was using T1 data. The sentence has been updated to clarify this, as suggested. We have narrowed the range for the colorbar, as suggested. The updated figure will be included with the revised manuscript.

**Comments on the dataset**

I would like to acknowledge the efforts put into the documentation of the data. The Readme files are extremely clear and this is of great help to a new user of the dataset. Below is a list of minor issues:

- there are some discrepancies in the naming of the files inside the readme files (for example in the SJ_FS_FLUX_readme, l. 134 – 136)

**Response:** Thank you for noticing this inconsistency. We have corrected the erroneous file names listed in the readme files, and the new readme files will be uploaded at the conclusion of the review process.

- OTT Parsivel: I could not find values of the diameter and velocity bins

**Response:** The mid-values and spreads for each of the 32 diameter and 32 fall speed bins have been added to the disdrometer readme file. A sentence has been added to Section 3.1.4 to inform the reader of this.

- MRR-PRO: I may have missed but could not find values of the velocity bins of the spectra (if they are indeed missing, one possibility would be to include the bounds + velocity resolution in the readme file so the user can reconstruct this array).

**Response:** Please see our response to point #22 above.

- Macrophotography: Can the user find a scale for the pictures?

**Response:** Images of a ruler were taken to provide a reference scale for users. The filenames of these images have been added to the readme file.

A sentence has been added to Section 3.2 that reads: *"Images of a ruler placed across the pad for scale were taken intermittently, for example when the camera was removed/replaced on the stand. Users can assume the scale remains constant between these images (each image is 2 cm across). The scale images are listed in the macrophotography readme file."*

**Please note:** Any changes to the dataset will be completed and made public at the end of the peer review process.

Reviewer 3:

RC2: 'Comment on essd-2023-59', Anonymous Referee #2, 10 Jun 2023

This paper presents a hydrometeorological dataset collected during an experimental measurement campaign that ran from December 2020 to April 2021 in the upper St. John River Basin in New Brunswick, a province in eastern Canada. The dataset includes automatic measurements available over the entire period, as well as detailed measurements during intensive observation periods structured around the passage of winter storms. The measured variables are very diverse: hydrometeor photos, atmospheric profiles, turbulent flux measurements, precipitation intensity, etc.

First of all, I would like to congratulate the authors on their highly ambitious and impressive measurement campaign, carried out in the middle of a tough pandemic period. The subject under study, the characterization of the phase and amount of winter precipitation in the Canadian Maritimes, is extremely relevant for the development of more accurate weather forecasting models. In this sense, the dataset represents a significant contribution to the community. In itself, the paper is very well written and structured, and the figures and tables are of high quality. However, I do have a few comments aimed at clarifying certain gray areas in the article. I would also encourage authors to be more selective about the data they make available to the community.

**1) Data Quality**

I understand that there are logistical constraints to such a deployment and that the installation of equipment in a given location is always the result of compromises. I am also aware that a very large number of instruments had to be deployed. Looking at the photos, however, I notice that the immediate surroundings of some of the weather stations do not seem to be conducive to quality measurements. For example, in Figure 3a, the anemometer is right next to a cedar hedge, which is bound to interfere with its measurements. In Figure 3d, it looks like the ground has been cleared of snow (the site appears to be adjacent to a parking lot) - has the snow height measurement been affected? Are the cedars casting a shadow on the radiometer? Similarly, in Figure 4, the fact that the snow has been heavily disturbed by footprints will change the measured albedo values... Has the footprint of the EC sensor remained intact, or has it also been affected by this perturbation (which will change its surface roughness)?

A good example of the kind of problem this raises is illustrated by the analysis of the data taken by the surface temperature sensor (l.316-327), which has been disturbed by shading effects, resulting from an improper installation. I believe it is better to have no data at all on a particular variable than to have data heavily contaminated by local "artifacts" associated with poor location or installation. Therefore, I encourage authors to be more selective in the observations they keep in the dataset. Please identify periods where artifacts specific to the immediate environment may have contaminated the dataset, and simply remove them from the overall dataset.

Similarly, soil sensors installed in frozen ground after the snow has been shoveled back on top of it (thus changing its structure) are, in my opinion, of little value to the study.

**Response:** Thank you for your constructive comments. To address the points raised in the first paragraph:

1. The wind data from the MUST Trailer have been removed from the dataset due to the proximity to the building, the cedar trees, and issues with extension of the mast.
2. Due to the requirement for snow removal around the area, snow depth at the MUST Trailer was measured by a sonic ranger and snowboard that was cleared after each storm. The procedure is detailed in Section 3.1.1.
3. The cedar trees are directly north of the line of instruments, and therefore do not shade the radiometers.
4. The image has been taken at the time of installation, and we agree this disturbance is non-trivial. We have added the following paragraph to the readme file for users to refer to: *"Staff approached the site of installation from the south to reduce disturbance of the fetch from the prevailing wind direction (north). Disturbance underneath and in the immediate vicinity of the tripod will have little effect on flux measurements as the footprint size is a function of horizontal wind speed and extends upwind of the instrument. However, we expect upward shortwave radiation values and albedo to be lower than expected (especially at low sun angles) until sufficient snowfall has occurred to preferentially fill in these surface indentations. An analysis of the timelapse images would assist this, as would the combination of radiation and albedo values with snowdepth measurements."*

In response to the second paragraph, we agree erroneous data are of little use in a finalized dataset, however, our motivation for archiving the SAJESS data presented here is to preserve the original raw data obtained during the campaign. This allows us (and other users) to evaluate the data without quality control mechanisms that could remove values indiscriminately, and reduce the potential for development of new quality control or correction methods. It also allows us to evaluate a variety of quality control processes available in the literature. Future installation of these instruments will also be improved by the ongoing evaluation of the raw SAJESS data, including the future use of automated data flags to indicate potentially erroneous values.

Our apologies if the manuscript indicates that these data are ready to be used as-is. A sentence has been added to the beginning of Section 4.2 that reads: *"To preserve the raw data recorded during SAJESS, no quality control related post-processing has been performed. We encourage all users to perform quality control and post-processing according to their needs."*

**2) Overview of the winter 2020-21**

It would be useful to present an overview of the winter of 2020-21 (number of precipitation events, liquid vs. solid fraction, temperature, etc.) and how it relates to the site's climatology. A table detailing **each storm would be highly relevant.**

**Response:** Thank you for the suggestion. We have revised Figure 7 and added text to the abstract and Sections 4.3.1, 4.3.2, and 4.3.3 to provide a broader picture of the conditions observed during

SAJESS.

For example, Section 4.3.1 has been renamed from *Winter 2020/21* to *Overview*, and now reads: *"Data covering 1 December 2020 – 30 April 2021 are provided by the instruments at the Fixed Station, which can be supplemented by the near-by ECCC station. Maximum snow depth measured at the Fixed Station was 65 cm, with snowfall amounts comparing well in timing, yet slightly less in magnitude (around 10%), than measured by the ECCC instruments (Figure 7). Maximum snow depth observed at 21 surrounding CoCoRaHS stations (Table 4) ranged from 50 cm to 79 cm, with an average of 54 cm. During this period, the ECCC Geonor measured a total of 287 mm of (liquid equivalent) precipitation, less than the climatological (1993-2022) mean of 364 mm.*

*During the IOP (8 March 2021 - 30 April 2021) manual observations were made during 13 storms (Fig. 6), for a total of 183 hours of precipitation-type observations. Observers were present for 93 hours of snow (8 storms), 63 hours of rain (5 storms), and 27 hours of mixed precipitation (more than one phase of precipitation occurring at the same time; 3 storms). Two storms (12 March 2021; 4-5 April 2021) included rain-snow transitions. Observers took a total of 4483 images of precipitation particles during the IOP. Using the ECCC Geonor, observers were present for 97 mm of the 105 mm of (liquid equivalent) precipitation that fell during the IOP period."*

An analysis of each storm is ongoing, and will be presented in a future manuscript.

**3) Federal Reference Station**

The ECCC station serves as the standard for measuring precipitation. We learn throughout the text that the precipitometer is a GEONOR, but it would be important to describe the type of shield, the exact model of the Geonor, as well as any other relevant equipment at the federal weather station. How to access the data should be explained. Would it even be possible to include it in the global **dataset?**

**Response:** We have now listed the equipment situated at the ECCC site in Section 2.2 by adding a sentence that reads: *"This site was chosen to allow for the SAJESS datasets to be supplemented and compared with records from the nearby ECCC station, which was comprised of a Geonor T-200B weighing precipitation gauge with a single Alter shield, three sonic ranger snow depth sensors, three Temperature/RH probes, and a RM Young wind monitor atop a 10-m mast."*

We have also added a sentence to the Data availability section that reads: *"Hourly and daily data from the Edmundston ECCC station are available via the ECCC Historical data webpage, with 1-minute raw data from the station available from the authors upon request. (https://climate.weather.gc.ca/historical_data/search_historic_data_e.html)."*

We have decided to not include the ECCC data in our datasets as these are (mostly) already available online.

**4) Turbulent flux data**

To be able to use the turbulent flux data, an analysis of the footprint is needed. We also need to know in detail how EasyFlux is set up. For instance, what kind of correction is used to rotate the coordinate system? Is there linear detrending? Etc.

**Response:** We agree that flux data can be complex, and while we wish to provide as much information as possible, we are also cautious of misinterpreting information that can already be found in material supplied by the manufacturer and referenced in the manuscript. For clarity, we include here a brief description of the processing steps detailed in the Easyflux manual, and the manual contains all equations and corrections complete with references. For brevity we have not included this in the manuscript but a sentence was added to refer the readers to the manual. If the reviewer still considers that it is necessary to add this information, we could add it in an Appendix.

*"Program Sequence of Measurement and Corrections*

*The main correction procedures and algorithms implemented into the program are listed below. For more information on the sequence of measurements and corrections, refer to Appendix I, EasyFlux DL CR6OP Process Flow Diagram (p. I-1). The appendices of this manual will give additional information on each major correction and its implementation in the program.*

*1. Despike and filter 10 Hz data using sonic and gas analyzer diagnostic codes, and signal strength and measurement output range thresholds.*

*2. Coordinate rotations with an option to use the double rotation method (Tanner and Thurtell 1969), or planar fit method (Wilczak et al. 2001).*

*3. Lag $CO_2$ and $H_2O$ measurements against sonic wind measurements for maximization of $CO_2$ and $H_2O$ fluxes (Horst and Lenschow 2009, Foken et al. 2012), with additional constraints to ensure lags are physically possible.*

*4. Frequency corrections using commonly used cospectra (Moore 1986, van Dijk 2002a, Moncrieff et al. 1997) and transfer functions of block averaging (Kaimal et al. 1989), line/volume averaging (Moore 1986, Moncrieff et al. 1997, Foken et al. 2012, van Dijk 2002a), time constants (Montgomery 1947, Shapland et al. 2014, Geankoplis 1993), and sensor separation (Horst and Lenschow 2009, Foken et al. 2012).*

*5. A modified SND correction (Schotanus et al. 1983) to derive sensible heat flux from sonic sensible heat flux following the implementation as outlined in van Dijk 2002b. Additionally, fully corrected real EasyFlux® DL CR6OP 46 sensible heat flux computed from fine-wire thermometry may be provided.*

*6. Correction for air density changes using WPL equations (Webb et al. 1980).*

*7. Data quality qualifications based on steady state conditions, surface layer turbulence characteristics, and wind directions following Foken et al. 2012 (or Foken et al. 2004 for the Flux_AmeriFluxFormat output table).*

*8. If energy balance sensors are used, calculation of energy closure based on energy balance measurements and corrected sensible and latent heat fluxes.*

*9. Footprint characteristics are computed using Kljun et al (2004) and Kormann and Meixner (2001)."*

Please refer to the EasyFlux manual for the references listed above.

**Specific comments**

L114 - Remove (a).

**Response**: This has been removed as suggested.

L180 - The hotplate has been tested, great. What was the outcome?

**Response:** We used the unheated Geonor weighing precipitation gauge at the nearby ECCC station as reference throughout SAJESS. Additional comparisons have also been made with data from the disdrometer and micro rain radar. We discuss the comparison of Hotplate performance with the Geonor in Section 4.2.3, which reads:

*"Similar to Thériault et al. (2021b), we compared 30-minute cumulative sums of the hotplate 1-minute accumulation for Dec 2020 – March 2021, with corrected Geonor data (using Kochendorfer et al., 2017) from the Edmundston ECCC station. These align well with the best results (reduced bias and RMSE) found for rain (>2°C) and snow (<-2°C). Finally, hotplate T1 (1-minute average) precipitation data are more sensitive and therefore better at reproducing higher precipitation rates than the T5 (5-minute average) data, resulting in a positive bias up to ~1.5 mm h$^{-1}$ (at 4-5 mm h$^{-1}$)."*

L346 – $CO_2$, $H_2O$

**Response:** This has been changed as suggested.

Figure 7 - Use three subplots: Temperature, Snow depth and Precip rate.

**Response:** Figure 7 has been updated to display three subplots as suggested.

Please note: Any changes to the dataset will be completed and made public at the end of the peer review process.

**Reviewer 4:**

**RC3: ['Comment on essd-2023-59'](), Robert Hellstrom, 07 Jul 2023**

General Comments:

As a data contribution, the authors present an intensive meteorological field observation of approximately 7 weeks during March and April 2023 aimed at advancing understanding of hydrometeor in the upper St. Johns Basin with broader impacts to river hazards in eastern Canada and northern New England. The combination of traditional ground-based weather stations and upper air balloon soundings with less common micro-wave signal attenuation of GOES satellite with MMR, hot-plate precipitation instruments, optical disdrometer for size and fall rate of hydrometeors, automated high-speed precipitation photography for snow-flake characteristics, manual macrophotography to study mixed-phase events, and integration of the community through citizen science make this a valuable contribution for potentially novel approaches for advancing scientific understanding of winter precipitation processes that pose significant hazards to communities, power production and river navigation. The quality of writing and organization of the manuscript are acceptable and easy to follow, describing the field site and surroundings, function of instrumentation, and limitations of instruments, but the manuscript could be improved by minor revisions that include more detailed descriptions of how the less conventional measurements are useful for precipitation studies, both to advance process-based **understanding and for operational meteorologists who forecast hazards.**

**Response:** Thank you for the suggestions. With comments from all four reviewers we have revised the Summary section of the manuscript, which now includes discussion about the potential use of SAJESS data from all instruments, including the less-conventional, as suggested. This revision also addresses two further comments mentioned below.

The Summary now reads:

[revised manuscript text omitted]

**Specific Comments by preprint line number:**

29-31 Abstract: This is a very important part of a data paper and deserves more elaboration in the manuscript

**Response:** The data management plan is now included with the readme documents of the

dataset. Minimal post-processing of the data has been performed. Archived raw data can be processed by users and we have provided suggestions for processing in the text.

43-47 Fig. 1: Legend, black dot for temperatures, are these the HOBO T/RH sensors set up by citizen scientists? Also, where are the CoCoRaHS gauges set up by students, important for low-cost measurements of spatial variability of precipitation in upper St. John's basin?

**Response:** The Temperature/RH sensors were distributed to willing community volunteers, with instructions on how to install the devices. Correct installation was confirmed by a (graduate student) member of the SAJESS team soon after initial setup. Note that this was taking place during periods of restrictive public health measures that limited contact between volunteers and the SAJESS team. We have revised Figure 1 and Table 4 to include the locations of CoCoRaHS gauges, noting where Temp/RH sensors were co-located.

48-50 Hazard impact: This sentence drives the need for a combination of measurements including mixed-phase precip. and air temperature. Emphasize this fact when discussing each of the measurements (section 3) so that the reader can connect the instrument data to its application for improving forecasting of hazards.

**Response:** Thank you for the suggestion. As section 3 is more focused on the instrumentation (although these obviously goes hand-in-hand with measurements) we have reserved the discussion of the use of SAJESS measurements for the Summary, as mentioned above.

61-63 Objective: I agree, but also data papers should describe _briefly_ how each observation could be useful to potential end-users (scientists, operational meteorologists, engineers, citizens, educators, FEMA, etc..). This would help broaden the audience.

**Response:** Thank you for the suggestion. Please see the revised Summary above for descriptions of how the dataset can be useful to potential user groups.

Additionally, the final paragraph of Section 1, where we state the objective of the paper, has been revised to now read: *"The SAJESS dataset contains meteorological and precipitation data that were collected at a fixed station from 1 December 2020 until 30 April 2021, and an intensive observation period (IOP) that took place from 8 March to 30 April 2021. The objective of this paper is to describe the data collected during SAJESS, provide examples of the measurements and how they can be combined to broaden the picture of meteorological conditions observed, and to illustrate to stakeholders and potential users the importance of the field campaign and dataset.*

*The paper is organized as follows: the sites used during SAJESS are described in Section 2, the instruments, manual observations, and other sources of data are detailed in Section 3, an overview of data processing, management, and validity, along with examples of data collected throughout SAJESS are presented in Section 4, and a summary, including a discussion on potential future analyses, is given in Section 5."*

74-78 Community volunteers: This is unusual and potentially high impact on the education of young scientists and the inclusion of community members to fill gaps and improve relations between ordinary citizens and stakeholders of the river system. This is unique and deserves attention in the summary at the end.

**Response:** Thank you for the suggestion. Please see the revised Summary above.

83 Figs. 1 and 2 and also 3: It would be helpful to provide an aerial photo of the sites to assess environmental surroundings and perhaps indicate prevailing storm tracks as overlay.

**Response:** We have revised Figure 1 to include annotated google earth images of the sites as suggested. The improved figure will be included in the revised manuscript.

125 SRS: Are there photos of these installations? Maybe one to show the setup.

**Response**: Thank you for your suggestion, but we have now clarified what instrumentation was used at each SRS site, Our original description may have sounded very technical. To an onlooker, the only equipment that is visible is a small box placed inside the residence that connects inline to the satellite dish coaxial cable. The processor inside the box is then configured to connect to the local wifi network. To reduce signal noise it is preferable that the satellite dish no longer be in service. These details have been included in Section 3.4.

152-156: Snowboard installed under the depth sounder is unconventional and may include several snowfall events if unattended. Of course, it would have been ideal to leave this undisturbed. The primary purpose of study was not snow depth, so perhaps make this argument.

**Response:** Daily and sub-daily clearing of snow boards is a common method for snowfall measurement in ski areas and avalanche forecasting operations. We have clarified the procedure that was undertaken for clearing the snow boards by adding a sentence that reads: *"The clearing of the snow boards was consistently performed by the volunteers/students. They were cleared at the beginning of observations before precipitation started, then cleared again once precipitation ended. Through these actions, the ensuing data should be considered by users as a snowfall measurement rather than true snow depth."*

157-159: IR temp is angled, make the argument for why, because oblique is a more accurate representation of vegetation canopy temps. This is useful only when grass is protruding through the snow surface, which tends to accelerate melt due to the lowering of albedo. Maybe make this argument, since IR thermometry is unconventional for a smooth snow surface and likely inaccurate. Snowpack temp. is typically measured with a thermistor string that also includes the first 10 to 40 cm of the substrate.

**Response:** The IR thermometers were angled (east and west) to avoid including the south facing

tripod leg being in the view of the sensors. It was envisioned that the temperature range between the two sensors would be small, and an average of the two readings would suffice. The larger than expected range of temperature between the two IR thermometers means that an average of the two would not be ideal. We present these details in Section 4.2.1. The deployment of a thermistor string was not in the scope of this experiment, however, we will consider it for future deployments.

161: Nice to see a direct flux system was available.  However, flux data are complex and the authors should explain why those data for projecting snowmelt.  Again, relating measurement to the objective of the dataset/paper.

**Response:** We agree with the reviewer that flux data are complex. The flux system was installed to complement measurements of snowpack evolution during the winter, and particularly, the melt season. Examples of the use of data from the flux instruments that we are investigating include estimations of sublimation from the snowpack, and radiation inputs to the snowpack during melt periods. We also plan to investigate patterns of energy and mass fluxes during the melt season.

177: Pond Engineering hotplate: Seems like a promising relatively low-cost approach to solid precip measurements.  Was there a heated rain gauge to validate/compare?

**Response:** We used the unheated Geonor weighing precipitation gauge at the nearby ECCC station as reference throughout SAJESS. Additional comparison has also included the data from the disdrometer and micro rain radar. We discuss the comparison of Hotplate performance with the GEONOR in Section 4.2.3, which reads: "Similar to Thériault et al. (2021b), we compared 30-minute cumulative sums of the hotplate 1-minute accumulation for December 2020 – March 2021, with corrected Geonor data (using Kochendorfer et al., 2017) from the Edmundston ECCC station. These align well with the best results (reduced bias and RMSE) found for rain (>2°C) and snow (<-2°C). Finally, the finer temporal resolution of the hotplate T1 (1-minute average) measurements are better at reproducing higher precipitation rates than the T5 (5-minute average) data, resulting in a positive bias up to ~1.5 mm h$^{-1}$ (when evaluated at 4-5 mm h$^{-1}$)."

185: Disdrometer: given the advancements in laser-optics tech.  The location should be shown in Fig. 3.

**Response:** Our apologies for any confusion, but there is no optical-disdrometer in Figure 3. The disdrometer is labelled in Figure 2.

189-191: What is the 32x32 matrix orientation, horizontal or vertical, for fall speed?

**Response:** We have clarified that the disdrometer data includes 1024 columns that represent the 32x32 matrix of particle fall speed (V) and diameter (D) bins. This 1024-long vector is formatted as: {V0D0, V0D1, V0D2… …V0D32, V1D0, V1D2… … V32D30, V32D31, V32D32}.

When reshaped to a 32x32 matrix it represents particle fall speed on the y-axis and diameter on

the x-axis, as seen below.

[Figure]

*Figure 1. A sample figure of 32 x 32 bin matrix of particle fall speed (y-axis) and diameter (x-axis) from the Parsivel2 disdrometer. To improve clarity, not all 32 bins are included in the limit of each axis.*

195-198: Good points about the need for keeping radar free of snow, a big challenge for data quality. Maybe consider a mechanical wiper triggered by snowfall...power is a limiting factor.

**Response:** Our experience is that the built-in heaters do well at preventing snow and ice build up, but we agree that power is sometimes a limiting factor. In another study we have been able to run the MRR (without heater) during warmer months on solar power, but this is impractical during winter. Perhaps a mechanical wiper will be the solution!

199: Snowflake camera is unique, but make the argument for its purpose in this study, how can the data be applied?

**Response:** Primarily, the MASC can be used during this study two ways: i) Images have been used individually to illustrate the hydrometeors observed, and highlight the riming conditions, crystal habit, size of particles observed at the surface. ii) Images will be post-processed using machine-learning algorithms to once again determine riming conditions and classify hydrometeors, however this time using a large number of images to increase the reliability of the results. Secondly, this was an experimental use of the MASC with regards to its deployment in mixed-phase precipitation events, and analysis of its performance is ongoing. Initial results indicate that particle fall speed and size estimations (similar to disdrometer results) from post-processing of the MASC images align well with the observed precipitation type, despite the images being 'less than ideal' from an aesthetic point of view.

We have added justification for the MASC to Section 3.1.6, and we now describe the experimental use of the MASC in Section 4.3.3.

208: MUST trailer: It's a shed for shelter that happens to have a wind sensor with faulty

extension, so perhaps leave the wind out and use the primary weather station.

**Response:** Yes, we agree. The wind data have now been removed from the repository, and the paragraph now reads *"Due to air leakage, the mast did not always maintain its full extension and therefore required re-extending at times. This, in addition to the proximity of the trailer to trees and a nearby structure has resulted in us excluding the wind data from the publicly available dataset."* We also emphasize the 'observational' aspect of the MUST trailer, rather than instrumentation that is the primary purpose of the Fixed Station in Section 2.3.

216: Macrophotography and time-lapse: Requires manual observation and it should be emphasized the importance of having human observers and training of students. Discuss the reliability and usefulness of the time-lapse photographs.

**Response:** Thank you for your suggestion, we have added to Section 3.2 and it now reads:

*"Observers were present at the MUST Trailer during periods of precipitation to report weather conditions, and to obtain macrophotographs of solid hydrometeors (Gibson and Stewart, 2007; Joe et al., 2014; Thériault et al., 2018; Lachapelle and Thériault, 2021a). This provided a running-record of the weather conditions during storm events and allowed for the field-training of students in manual observations, and identification of precipitation types and snow crystal habits. The recording of manual observations can also remove doubt about precipitation arriving at the surface. This is especially important with respect to hydrometeor type during periods of near-freezing conditions, during which a mixture or changes in precipitation phase or type can occur."*

We also clarify the importance of time lapse images with a sentence that reads: *"Hourly images of each site, including the surface conditions around the instruments, were captured by a time lapse camera. Similar previous campaigns by the authors have found time-lapse images to be very useful in confirming sky and surface conditions such as snow-on-the-ground onset and they have also provided evidence of wildlife encounters with the instrumentation. Precipitation type diagnoses, however, are not usually possible with these images."*

226: Upper Air: the iMet system seems very reliable and cost-effective. Please discuss briefly the criteria for making 52 observations. Were the sounding instruments retrievable?

**Response:** Thank you for your comment. The paragraph has been improved to clarify the timings of balloon releases, and now reads: *"Soundings were timed to coincide with the standard synoptic times of 00, 03, 06, 09, 12, 15, 18, or 21 UTC. Some additional launches were attempted to coincide with precipitation phase transitions."* We also now clarify that sonde retrieval was not undertaken during this experiment.

232: SRS: The system was designed for liquid precip. Discuss briefly the potential for mixed

precip. given partial liquid content.

**Response:** Initial results from the SRS system deployed during SAJESS appear promising for the use of these devices in mixed-phase precipitation events. To clarify that SAJESS provided an experimental opportunity for the use of the SRS system, we have reworded Section 3.4 to read: *"SAJESS provided an experimental opportunity for SRS devices to monitor the liquid content in cases of mixed precipitation and wet (melting) snow. The SRS system tested in Edmundston was composed of a set of distributed SML sensors, as described by Colli et al. (2019), connected to a central processing and analysis node to reconstruct the bi-dimensional rainfall field in real time. To an onlooker, the only equipment visible is a small box placed inside the residence that connects inline to the satellite dish coaxial cable. The processor inside the box is then configured to connect to the local Wi-Fi network. To reduce signal noise, it is preferable that the satellite dish no longer be in service. These results will be published once work is complete."*

247: CoCoRaHS: Emphasize the importance of connecting with the community and the potential for encouraging younger students to get involved. Stress the importance of low-cost distributed precip. networks for assessing spatial variability.

**Response:** Thank you for your suggestion. We have added a short sentence that reads: *"This provided an opportunity for students to engage in the data collection process, and to learn about the importance (and difficulties) of precipitation determination."*

Furthermore, the topic of engagement of community volunteers during SAJESS forms the basis of a manuscript in preparation where we will expand on the importance of citizen science and low-cost distributed networks, as suggested.

257: In Fig. 1 these are marked as temperature sensors, not T/RH combo. Humidity and dew point are very important components of precipitation and should be included. Also, perhaps stress the importance of low-cost distributed networks for assessing spatial variations in T/RH.

**Response:** Thank you for noticing this inconsistency. Figure 1 and Table 4 have been updated to clarify that the sensors measure temperature and relative humidity, and to also include the locations of the CoCoRaHS stations.
Section 3.5.2 also now includes the following sentence: *"These low-cost, robust sensors provided a broad (50-60 km) network to assess spatial variability in near-surface temperature/RH, especially during the passage of fronts and the onset/cessation of precipitation."*

Section 4: This section is well-written and provides adequate details of validation and limitations of the measurements.

**Response:** Thank you.

401: Examples of data and obs: It would be nice to loop back to the purpose of the study for each of the measurements taken, perhaps one or two sentences on how each could be used to

improve river forecasting or advance understanding of the processes.

**Response:** Thank you for the suggestion. We believe the revision of the Summary (see above) addresses this point.

457: section 4.4: This is a good summary of the limitations of the study.

**Response:** Thank you.

Section 5: Summary is very brief and makes good points, but explaining how the most reliable observations add to the understanding of hydrometeors and potential for better forecasting would be useful.

**Response:** Thank you for the suggestion. Please see the response above that contains the revised Summary.

Please note: Any changes to the dataset will be completed and made public at the end of the peer review process.

---

## Author Response (AR2)

**Response to Topic editor decision: Publish subject to minor revisions (review by editor) 1 Oct 2023, by David Carlson**

Interesting data set on winter precip variables. Very good technical responses to reviewer comments. Substantial truncations and non-seqs remain, particularly in abstract. I do not want to inflict those on proof readers; these researchers will know better if and how to correct. Please use native English speaker/reader to evaluate and comment? Return abstract only if desired.

Small easy changes, quick. Thank you for using ESSD.

**Response:** Our sincere apologies for the poorly revised abstract. Please find the new version below, as well as uploaded to the 'abstract' section of the file uploads page.

**Abstract.** The amount and phase of cold season precipitation accumulating in the upper Saint John River (SJR) basin are critical factors in determining spring runoff, ice-jams, and flooding. To study the impact of winter and spring storms on the snowpack in the upper SJR basin, the Saint John River Experiment on Cold Season Storms (SAJESS) was conducted during winter/spring 2020-21. Here, we provide an overview of the SAJESS study area, field campaign, and data collected. The upper SJR basin represents 41% of the entire SJR watershed and encompasses parts of the US state of Maine and the Canadian provinces of Quebec and New Brunswick. In early December 2020, meteorological instruments were co-located with an Environment and Climate Change Canada station near Edmundston, New Brunswick. This included a separate weather station for measuring standard meteorological variables, an optical disdrometer, and a micro rain radar. This instrumentation was augmented during an intensive observation period that also included upper-air soundings, surface weather observations, a multi-angle snowflake camera, and macrophotography of solid hydrometeors throughout March and April 2021. During the study, the region experienced a lower-than-average snowpack that peaked at ~65 cm, with a total of 287 mm of precipitation (liquid equivalent) falling between December 2020 and April 2021, a 21% lower amount of precipitation than the climatological normal. Observers were present for 13 storms during which they conducted 183 hours of precipitation observations and took more than 4000 images of hydrometeors. The inclusion of local volunteers and schools provided an additional 1700 measurements of precipitation amounts across the area.

The resulting datasets are publicly available from the Federated Research Data Repository at https://doi.org/10.20383/103.0591 (Thompson et al., 2023). We also include a synopsis of the data management plan, and a brief assessment of the rewards and challenges of conducting the field campaign and utilizing community volunteers for citizen science.